# Learning Better Certified Models from Empirically-Robust Teachers

## Abstract

Adversarial training attains strong empirical robustness to specific adversarial attacks by training on concrete adversarial perturbations, but it produces neural networks that are not amenable to strong robustness certificates through neural network verification. On the other hand, earlier certified training schemes directly train on bounds from network relaxations to obtain models that are certifiably robust, but display sub-par standard performance. Recent work has shown that state-of-the-art trade-offs between certified robustness and standard performance can be obtained through a family of losses combining adversarial outputs and neural network bounds. Nevertheless, differently from empirical robustness, verifiability still comes at a significant cost in standard performance. In this work, we propose to leverage empirically-robust teachers to improve the performance of certifiably-robust models through knowledge distillation. Using a versatile feature-space distillation objective, we show that distillation from adversarially-trained teachers consistently improves on the state-of-the-art in certified training for ReLU networks across a series of robust computer vision benchmarks.

## 1 Introduction

Deep learning systems deployed in safety-critical applications must be robust to adversarial examples (Biggio et al., 2013; Szegedy et al., 2014; Goodfellow et al., 2015): imperceptible input perturbations that induce unintended behaviors such as misclassifications. Formal robustness certificates can be obtained through neural network verification algorithms (Tjeng et al., 2019; Lomuscio & Maganti, 2017; Ehlers, 2017). However, these techniques have a worst-case runtime that is exponential in network size even on piecewise-linear models (Katz et al., 2017). While *empirical* robustness to specific adversarial attacks can be attained by training against adversarial inputs (Madry et al., 2018), a technique known as adversarial training, neural network verifiers fail to provide robustness certificates on the resulting networks in a reasonable time. This is in spite of sustained progress in verification algorithms, which now couple specialized network convex relaxations (Zhang et al., 2018; Xu et al., 2021; Wong & Kolter, 2018; De Palma et al., 2024a) with efficient divide and conquer strategies (Bunel et al., 2020b; Henriksen & Lomuscio, 2021) within hardware-accelerated branch-and-bound frameworks (Wang et al., 2021; Ferrari et al., 2022; De Palma et al., 2021).

Networks amenable to robustness certificates through neural network verification can be obtained using so-called certified training schemes. Earlier approaches (Wong & Kolter, 2018; Zhang et al., 2020; Gowal et al., 2019; Mirman et al., 2018) proposed to train networks by computing the loss on neural network bounds obtained from convex relaxations, effectively deploying a building block of network verifiers within the training loop. Counter-intuitively, and owing to their relative smoothness and continuity, the loosest relaxations were found to outperform the others in this context (Jovanović et al., 2022; Lee et al., 2021). While the resulting networks enjoy strong verifiability using the same relaxations employed at training time, this is achieved at a significant cost in standard performance. Relying on branch-and-bound frameworks to perform post-training verification, a more recent line of works has shown that better trade-offs between certified robustness and standard performance can be obtained by combining methods based on convex relaxations with adversarial training (Balunovic & Vechev, 2020; De Palma et al., 2022; Müller et al., 2023; Mao et al., 2023). This was then formalized into the notion of *expressivity* (De Palma et al., 2024b), which entails the ability of a certified training loss to span a continuous range of trade-offs between pure adversarial training and the earlier losses based on convex relaxations, and can be easily implemented through convex combinations.

While, as attested by a recent study (Mao et al., 2025), networks trained through expressive losses produce state-of-the-art certifiably-robust models, their standard performance is still far from ideal. Noting that empirically-robust models display significantly better standard performance than certifiably-robust models (Croce et al., 2021), we believe they could be directly employed to improve the certified training process. Specifically, we aim to produce better certifiably-robust models by performing knowledge distillation (Hinton et al., 2015; Romero et al., 2015) from an empirically-robust teacher to the target model, leading to the following contributions:

- We introduce a novel and versatile feature-space distillation loss, which can transfer the knowledge of a teacher onto any convex combination between adversarial student features and bounds from its convex relaxations (§3.2).

- We tightly couple the proposed distillation objective with an existing expressive certified training loss, calling the CC-Dist the resulting algorithm (§3.3). We show that CC-Dist can successfully learn from an adversarially-trained teacher while at the same time significantly surpassing it in terms of certified robustness (§5.2).

- We present a comprehensive experimental evaluation of CC-Dist across medium and larger-scale vision benchmarks from the certified training literature, and show that the novel distillation loss enhances both standard performance and certified robustness across all considered benchmarks (§5.1). In particular, CC-Dist attains a new state-of-the-art for ReLU architectures on all setups, with significant improvements upon results from the literature on TinyImageNet and downscaled Imagenet.

We believe our work to be a further step towards bridging the ever-present gap between certified and empirical adversarial robustness. Code for CC-Dist is provided as part of the supplementary material.

## 2 BACKGROUND

Let lowercase letters denote scalars (e.g., $a \in \mathbb{R}$) boldface lowercase letters denote vectors (e.g., $\mathbf{a} \in \mathbb{R}^n$), uppercase letters denote matrices (e.g., $A \in \mathbb{R}^{n \times m}$), calligraphic letters denote sets (e.g. $\mathcal{A} \subset \mathbb{R}^n$) and brackets denote intervals (e.g., $[\mathbf{a}, \mathbf{b}]$). Furthermore, let lowercase single-letter subscripts denote vector indices (for instance, $\mathbf{a}_i$, or $f(\mathbf{a})_i$ for a vector-valued function $f$), and let the vector of all entries of $\mathbf{a}$ except its $i$-th entry be denoted by $\mathbf{a}_{\bar{i}}$. We will write $\mathbf{1}^n \in \mathbb{R}^n$ for the unit vector, $I^n \in \mathbb{R}^{n \times n}$ for the identity matrix, and $_j I^n \in \mathbb{R}^{n \times n}$ for a matrix whose $j$-th column is the unit vector and filled with zeros otherwise. Abusing notation, given a vector-valued objective function $a(\mathbf{x})$ we will denote by $\min_{\mathbf{x} \in \mathcal{A}} a(\mathbf{x})$ the vector storing the minimum of $\min_{\mathbf{x} \in \mathcal{A}} a(\mathbf{x})_i$ in its $i$-th entry. Finally, we use the following shorthands: $[A]_+ := \max\{0, A\}$, $[A]_- := \min\{0, A\}$, $[\![a, b]\!] := \{a, a+1, \ldots, b\}$. Appendix F provides further details on the employed notation.

Let $(\mathbf{x}, y) \sim \mathcal{D}$ be a $k$-way classification dataset with points $\mathbf{x} \in \mathbb{R}^d$ and labels $y \in [\![1, k]\!]$. We aim to train a feed-forward ReLU neural network $f_{\boldsymbol{\theta}} : \mathbb{R}^d \to \mathbb{R}^k$ with parameters $\boldsymbol{\theta} \in \mathbb{R}^p$ such that each point $\mathbf{x}$ from $\mathcal{D}$ and the allowed adversarial perturbations $\mathcal{C}_\epsilon(\mathbf{x}) := \{\mathbf{x}' : \|\mathbf{x}' - \mathbf{x}\|_\infty \le \epsilon\}$ around it are correctly classified:

$$\mathbf{x}' \in \mathcal{C}_\epsilon(\mathbf{x}) \implies \operatorname*{argmax}_i f_{\boldsymbol{\theta}}(\mathbf{x})_i = y. \tag{1}$$

### 2.1 NEURAL NETWORK VERIFICATION

Neural network verification is used to check whether equation (1) holds on a given network $f_{\boldsymbol{\theta}}$, providing a deterministic robustness certificate. Let $\mathbf{z}_{f_{\boldsymbol{\theta}}}(\mathbf{x}, y)$ denote the differences between the ground truth logits and the other logits: $\mathbf{z}_{f_{\boldsymbol{\theta}}}(\mathbf{x}, y) := (f_{\boldsymbol{\theta}}(\mathbf{x})_y \mathbf{1}^k - f_{\boldsymbol{\theta}}(\mathbf{x}))$. Verifiers solve the following optimization problem:

$$\mathbf{z}_{f_{\boldsymbol{\theta}}}^{\mathcal{C}_\epsilon}(\mathbf{x}, y) := \min_{\mathbf{x}' \in \mathcal{C}_\epsilon(\mathbf{x})} \mathbf{z}_{f_{\boldsymbol{\theta}}}(\mathbf{x}', y). \tag{2}$$

If all entries of $\mathbf{z}_{f_{\boldsymbol{\theta}}}^{\mathcal{C}_\epsilon}(\mathbf{x}, y)_{\bar{y}}$ are positive, then equation (1) holds, implying that the network is guaranteed to be robust to any given attack in $\mathcal{C}_\epsilon(\mathbf{x})$. An algorithm computing $\mathbf{z}_{f_{\boldsymbol{\theta}}}^{\mathcal{C}_\epsilon}(\mathbf{x}, y)$ exactly is called a complete verifier: this is typically done through branch-and-bound (Bunel et al., 2018; De Palma et al., 2024a; Wang et al., 2021; Ferrari et al., 2022; Henriksen & Lomuscio, 2021). As this was shown to be NP-complete (Katz et al., 2017), a series of algorithms propose to compute

less expensive lower bounds $\underline{\mathbf{z}}_{f_{\boldsymbol{\theta}}}^{\mathcal{C}_\epsilon}(\mathbf{x}, y) \leq \mathbf{z}_{f_{\boldsymbol{\theta}}}^{\mathcal{C}_\epsilon}(\mathbf{x}, y)$ (Zhang et al., 2018; Wong & Kolter, 2018; Dvijotham et al., 2018; Singh et al., 2019) by operating on network relaxations (incomplete verifiers), with $\underline{\mathbf{z}}_{f_{\boldsymbol{\theta}}}^{\mathcal{C}_\epsilon}(\mathbf{x}, y)_{\bar{y}} > 0$ successfully providing a robustness certificate. The least expensive incomplete verifier is called Interval Bound Propagation (IBP) (Gowal et al., 2019; Mirman et al., 2018), which is obtained by applying interval arithmetics (Sunaga, 1958; Hickey et al., 2001) to the network operators. We refer the reader to appendix A for further details.

## 2.2 Certified training

In principle, a network can be trained for verified robustness (certified training) by replacing the employed classification loss $\mathcal{L} : \mathbb{R}^k \times [\![1, k]\!] \to \mathbb{R}$ (e.g., cross-entropy) with its worst-case across the adversarial perturbations (Madry et al., 2018):

$$\mathcal{L}_{f_{\boldsymbol{\theta}}}^{\mathcal{C}_\epsilon}(\mathbf{x}, y) := \max_{\mathbf{x}' \in \mathcal{C}_\epsilon(\mathbf{x})} \mathcal{L}(f_{\boldsymbol{\theta}}(\mathbf{x}'), y). \tag{3}$$

However, computing $\mathcal{L}_{f_{\boldsymbol{\theta}}}^{\mathcal{C}_\epsilon}(\mathbf{x}, y)$ exactly is as hard as solving problem (2), and is hence typically replaced by an approximation. Let $\mathbf{x}_{\text{adv}}$ denote the output of an adversarial attack, for instance PGD (Madry et al., 2018), on the network $f_{\boldsymbol{\theta}}$. Lower bounds to $\mathcal{L}_{f_{\boldsymbol{\theta}}}^{\mathcal{C}_\epsilon}(\mathbf{x}, y)$ can be obtained by evaluating the loss at $\mathbf{x}_{\text{adv}}$:

$$\underline{\mathcal{L}}_{f_{\boldsymbol{\theta}}}^{\mathcal{C}_\epsilon}(\mathbf{x}, y) := \mathcal{L}(f_{\boldsymbol{\theta}}(\mathbf{x}_{\text{adv}}), y) \leq \mathcal{L}_{f_{\boldsymbol{\theta}}}^{\mathcal{C}_\epsilon}(\mathbf{x}, y). \tag{4}$$

Networks trained using $\mathcal{L}(f_{\boldsymbol{\theta}}(\mathbf{x}_{\text{adv}}), y)$ (adversarial training) typically enjoy strong empirical robustness to the same types of attacks employed during training, but are not amenable to formal guarantees, which are the focus of this work. Assume the loss is monotonically increasing with respect to the non-ground-truth network logits and that it is translation-invariant: $\mathcal{L}(-\mathbf{z}_{f_{\boldsymbol{\theta}}}(\mathbf{x}, y), y) = \mathcal{L}(f_{\boldsymbol{\theta}}(\mathbf{x}), y)$ (Wong & Kolter, 2018). This holds for common losses such as cross-entropy. Given bounds $\underline{\mathbf{z}}_{f_{\boldsymbol{\theta}}}^{\mathcal{C}_\epsilon}(\mathbf{x}, y)$ from an incomplete verifier, and setting $\underline{\mathbf{z}}_{f_{\boldsymbol{\theta}}}^{\mathcal{C}_\epsilon}(\mathbf{x}, y)_y = 0$, an upper bound to $\mathcal{L}_{f_{\boldsymbol{\theta}}}^{\mathcal{C}_\epsilon}(\mathbf{x}, y)$ can then be obtained as:

$$\bar{\mathcal{L}}_{f_{\boldsymbol{\theta}}}^{\mathcal{C}_\epsilon}(\mathbf{x}, y) := \mathcal{L}(-\underline{\mathbf{z}}_{f_{\boldsymbol{\theta}}}^{\mathcal{C}_\epsilon}(\mathbf{x}, y), y) \geq \mathcal{L}_{f_{\boldsymbol{\theta}}}^{\mathcal{C}_\epsilon}(\mathbf{x}, y). \tag{5}$$

While networks trained via $\mathcal{L}(-\underline{\mathbf{z}}_{f_{\boldsymbol{\theta}}}^{\mathcal{C}_\epsilon}(\mathbf{x}, y), y)$ display strong verifiability through the same incomplete verifiers used to compute the $\underline{\mathbf{z}}_{f_{\boldsymbol{\theta}}}^{\mathcal{C}_\epsilon}(\mathbf{x}, y)$ bounds, the most successful option being IBP (Jovanović et al., 2022), they display sub-par standard performance. More recent and successful methods rely on losses featuring a combination between adversarial attacks and bounds from incomplete verifiers (De Palma et al., 2022; Balunovic & Vechev, 2020; Mao et al., 2023; Müller et al., 2023), pairing better standard performance with stronger verifiability through branch-and-bound. In particular, the ability of a loss function to span a continuous range of trade-offs between lower and upper bounds to $\mathcal{L}_{f_{\boldsymbol{\theta}}}^{\mathcal{C}_\epsilon}(\mathbf{x}, y)$, termed *expressivity*, was found to be crucial to maximize performance (De Palma et al., 2024b). Nevertheless, the best-performing certifiably-robust models still display worse robustness-accuracy trade-offs than empirically-robust models (De Palma et al., 2024b; Croce et al., 2021).

## 2.3 Knowledge distillation

Knowledge distillation (Hinton et al., 2015) was introduced as a method to transfer the predictive ability of a large teacher model $t_{\boldsymbol{\theta}^t} : \mathbb{R}^d \to \mathbb{R}^k$ onto the target model $f_{\boldsymbol{\theta}}$, termed the student. Let $\text{KL}_T : \mathbb{R}^k \times \mathbb{R}^k \to \mathbb{R}$ denote a KL divergence incorporating a softmax operation with temperature $T$. In its standard form, knowledge distillation pairs the employed classification loss with a KL divergence term where the teacher logits, termed soft labels, act as target distribution:

$$\mathcal{L}_{f_{\boldsymbol{\theta}}}^{\text{KD}}(\lambda; \mathbf{x}, y) := \mathcal{L}(f_{\boldsymbol{\theta}}(\mathbf{x}), y) + T^2 \lambda \, \text{KL}_T(f_{\boldsymbol{\theta}}(\mathbf{x}), t_{\boldsymbol{\theta}^t}(\mathbf{x})). \tag{6}$$

In order to provide more granular teacher information to the student, a series of works perform knowledge distillation on the intermediate activations (Romero et al., 2015; Zagoruyko & Komodakis, 2017; Heo et al., 2019): this is referred to as feature-space distillation. Let us write both the teacher and the student as the composition of a classification head with a feature map: $t_{\boldsymbol{\theta}^t} := g_{\boldsymbol{\theta}_g^t}^t \circ h_{\boldsymbol{\theta}_h^t}^t$ and $f_{\boldsymbol{\theta}} := g_{\boldsymbol{\theta}_g} \circ h_{\boldsymbol{\theta}_h}$, respectively, where $\boldsymbol{\theta}^t = [\boldsymbol{\theta}_h^t, \boldsymbol{\theta}_g^t]^T$ and $\boldsymbol{\theta} = [\boldsymbol{\theta}_h, \boldsymbol{\theta}_g]^T$. In its simplest form,

when the feature spaces of the teacher and student share the same dimensionality $w$, feature-space distillation encourages similarity between teacher and student features through a squared $\ell_2$ term:

$$\mathcal{L}_{f_{\boldsymbol{\theta}}}^{\text{F-KD}}(\lambda; \mathbf{x}, y) := \mathcal{L}(f_{\boldsymbol{\theta}}(\mathbf{x}), y) + \lambda \left\| h_{\boldsymbol{\theta}_h}(\mathbf{x}) - h_{\boldsymbol{\theta}_h^t}^t(\mathbf{x}) \right\|_2^2.$$

Both $\mathcal{L}_{f_{\boldsymbol{\theta}}}^{\text{KD}}(\lambda; \mathbf{x}, y)$ and $\mathcal{L}_{f_{\boldsymbol{\theta}}}^{\text{F-KD}}(\lambda; \mathbf{x}, y)$ were originally designed to transfer standard network performance from teacher to student, and do not take robustness into account. Recent works have therefore focused on designing specialized distillation schemes that transfer either empirical adversarial robustness (Goldblum et al., 2020; Zhu et al., 2022; Zi et al., 2021; Muhammad et al., 2021) or probabilistic certified robustness (Vaishnavi et al., 2022) from teacher to student. Aiming to improve state-of-the-art trade-offs between deterministic certified robustness and standard performance, we will present a novel training scheme that transfers knowledge from an empirically-robust teacher through feature-space distillation.

# 3    KNOWLEDGE DISTILLATION FOR CERTIFIED ROBUSTNESS

We aim to leverage the empirical robustness of adversarially-trained networks to train better certifiably-robust models. Owing to its state-of-the-art certified training performance, we will couple an existing expressive loss function (§3.1) with a novel and versatile feature-space distillation term (§3.2). Pseudo-code and proofs of technical results are respectively provided in appendices C and B.

## 3.1    EXPRESSIVE LOSSES: CC-IBP

De Palma et al. (2024b) define a parametrized loss function $\mathcal{L}_{f_{\boldsymbol{\theta}}}^{\mathcal{C}_\epsilon}(\alpha; \mathbf{x}, y)$ to be expressive if: (i) $\mathcal{L}_{f_{\boldsymbol{\theta}}}^{\mathcal{C}_\epsilon}(0; \mathbf{x}, y) = \mathcal{L}_{f_{\boldsymbol{\theta}}}^{\mathcal{C}_\epsilon}(\mathbf{x}, y)$ and $\mathcal{L}_{f_{\boldsymbol{\theta}}}^{\mathcal{C}_\epsilon}(1; \mathbf{x}, y) = \bar{\mathcal{L}}_{f_{\boldsymbol{\theta}}}^{\mathcal{C}_\epsilon}(\mathbf{x}, y)$; (ii) $\mathcal{L}_{f_{\boldsymbol{\theta}}}^{\mathcal{C}_\epsilon}(\mathbf{x}, y) \leq \mathcal{L}_{f_{\boldsymbol{\theta}}}^{\mathcal{C}_\epsilon}(\alpha; \mathbf{x}, y) \leq \bar{\mathcal{L}}_{f_{\boldsymbol{\theta}}}^{\mathcal{C}_\epsilon}(\mathbf{x}, y)$ $\forall \, \alpha \in [0, 1]$; (iii) $\mathcal{L}_{f_{\boldsymbol{\theta}}}^{\mathcal{C}_\epsilon}(\alpha; \mathbf{x}, y)$ is continuous and monotonically increasing for $\alpha \in [0, 1]$.

As demonstrated by the empirical performance of three different expressive losses obtained through convex combinations between adversarial and incomplete verification terms, expressivity results in state-of-the-art certified training performance (De Palma et al., 2024b). We here focus on CC-IBP, which implements an expressive loss by evaluating $\mathcal{L}$ on convex combinations between adversarial logit differences $\mathbf{z}_{f_{\boldsymbol{\theta}}}(\mathbf{x}_{\text{adv}})$ and lower bounds $\underline{\mathbf{z}}_{f_{\boldsymbol{\theta}}}^{\mathcal{C}_\epsilon}(\mathbf{x}, y)$ to the logit differences computed via IBP:

$$^{\text{CC}}\mathcal{L}_{f_{\boldsymbol{\theta}}}^{\mathcal{C}_\epsilon}(\alpha; \mathbf{x}, y) := \mathcal{L}\left(-\,^{\text{CC}}\mathbf{z}_{f_{\boldsymbol{\theta}}}^{\mathcal{C}_\epsilon}(\alpha; \mathbf{x}, y), y\right),$$
$$\text{where:} \quad ^{\text{CC}}\mathbf{z}_{f_{\boldsymbol{\theta}}}^{\mathcal{C}_\epsilon}(\alpha; \mathbf{x}, y) := (1 - \alpha)\,\mathbf{z}_{f_{\boldsymbol{\theta}}}(\mathbf{x}_{\text{adv}}, y) + \alpha\,\underline{\mathbf{z}}_{f_{\boldsymbol{\theta}}}^{\mathcal{C}_\epsilon}(\mathbf{x}, y). \tag{7}$$

As we will show in §3.3, CC-IBP can be tightly coupled with a specialized distillation loss, which we present in the next subsection.

## 3.2    DISTILLATION LOSS

Certified training is concerned with worst-case network behavior. Mirroring the robust loss in §2.2, a worst-case feature-space distillation loss over the perturbations would take the following form:

$$\mathcal{R}_{f_{\boldsymbol{\theta}}}^{\mathcal{C}_\epsilon}(\mathbf{x}, y) := \max_{\mathbf{x}' \in \mathcal{C}_\epsilon(\mathbf{x})} \left\| h_{\boldsymbol{\theta}_h}(\mathbf{x}') - h_{\boldsymbol{\theta}_h^t}^t(\mathbf{x}) \right\|_2^2. \tag{8}$$

However, as for $\mathcal{L}_{f_{\boldsymbol{\theta}}}^{\mathcal{C}_\epsilon}(\mathbf{x}, y)$, computing $\mathcal{R}_{f_{\boldsymbol{\theta}}}^{\mathcal{C}_\epsilon}(\mathbf{x}, y)$ exactly amounts to solving a non-convex optimization problem over the features. As for equation (4), a lower bound $\underline{\mathcal{R}}_{f_{\boldsymbol{\theta}}}^{\mathcal{C}_\epsilon}(\mathbf{x}, y)$ can be obtained by evaluating the left-hand side of equation (8) at the adversarial input $\mathbf{x}_{\text{adv}}$:

$$\underline{\mathcal{R}}_{f_{\boldsymbol{\theta}}}^{\mathcal{C}_\epsilon}(\mathbf{x}, y) := \left\| h_{\boldsymbol{\theta}_h}(\mathbf{x}_{\text{adv}}) - h_{\boldsymbol{\theta}_h^t}^t(\mathbf{x}) \right\|_2^2 \leq \mathcal{R}_{f_{\boldsymbol{\theta}}}^{\mathcal{C}_\epsilon}(\mathbf{x}, y).$$

Similarly to $\bar{\mathcal{L}}_{f_{\boldsymbol{\theta}}}^{\mathcal{C}_\epsilon}(\mathbf{x}, y)$, an upper bound $\bar{\mathcal{R}}_{f_{\boldsymbol{\theta}}}^{\mathcal{C}_\epsilon}(\mathbf{x}, y)$ can be instead computed resorting to IBP.

**Proposition 3.1.** *Let $\underline{h}_{\boldsymbol{\theta}_h}^{\mathcal{C}_\epsilon}(\mathbf{x})$ and $\bar{h}_{\boldsymbol{\theta}_h}^{\mathcal{C}_\epsilon}(\mathbf{x})$ respectively denote IBP lower and upper bounds to the student features:*

$$\underline{h}_{\boldsymbol{\theta}_h}^{\mathcal{C}_\epsilon}(\mathbf{x}) \leq h_{\boldsymbol{\theta}_h}(\mathbf{x}') \leq \bar{h}_{\boldsymbol{\theta}_h}^{\mathcal{C}_\epsilon}(\mathbf{x}), \quad \forall \mathbf{x}' \in \mathcal{C}_\epsilon(\mathbf{x}).$$

*The loss function* $\bar{\mathcal{R}}_{f_{\boldsymbol{\theta}}}^{\mathcal{C}_\epsilon}(\mathbf{x}, y) := \sum_i \max \left\{ \left( \underline{h}_{\boldsymbol{\theta}_h}^{\mathcal{C}_\epsilon}(\mathbf{x})_i - h_{\boldsymbol{\theta}_h^t}^t(\mathbf{x})_i \right)^2, \left( \bar{h}_{\boldsymbol{\theta}_h}^{\mathcal{C}_\epsilon}(\mathbf{x})_i - h_{\boldsymbol{\theta}_h^t}^t(\mathbf{x})_i \right)^2 \right\}$ *is an upper bound to the worst-case distillation loss from equation* (8): $\bar{\mathcal{R}}_{f_{\boldsymbol{\theta}}}^{\mathcal{C}_\epsilon}(\mathbf{x}, y) \geq \mathcal{R}_{f_{\boldsymbol{\theta}}}^{\mathcal{C}_\epsilon}(\mathbf{x}, y)$.

In order to preserve the greatest degree of flexibility, and mirroring expressive losses (§3.1), we aim to design a parametrized feature-space distillation loss $^{\mathrm{CC}}\mathcal{R}_{f_{\boldsymbol{\theta}}}^{\mathcal{C}_\epsilon}(\alpha; \mathbf{x}, y)$ that can span a continuous range of trade-offs between $\underline{\mathcal{R}}_{f_{\boldsymbol{\theta}}}^{\mathcal{C}_\epsilon}(\mathbf{x}, y)$ and $\bar{\mathcal{R}}_{f_{\boldsymbol{\theta}}}^{\mathcal{C}_\epsilon}(\mathbf{x}, y)$. Let us denote by $^{\mathrm{CC}}\bar{h}_{\boldsymbol{\theta}_h}^{\mathcal{C}_\epsilon}(\alpha; \mathbf{x})$ and $^{\mathrm{CC}}\underline{h}_{\boldsymbol{\theta}_h}^{\mathcal{C}_\epsilon}(\alpha; \mathbf{x})$ convex combinations of the adversarial student features $h_{\boldsymbol{\theta}_h}(\mathbf{x}_{\mathrm{adv}})$ with $\bar{h}_{\boldsymbol{\theta}_h}^{\mathcal{C}_\epsilon}(\mathbf{x})$ and $\underline{h}_{\boldsymbol{\theta}_h}^{\mathcal{C}_\epsilon}(\mathbf{x})$, respectively:

$$
\begin{aligned}
{}^{\mathrm{CC}}\bar{h}_{\boldsymbol{\theta}_h}^{\mathcal{C}_\epsilon}(\alpha; \mathbf{x}) &:= (1 - \alpha) \, h_{\boldsymbol{\theta}_h}(\mathbf{x}_{\mathrm{adv}}) + \alpha \, \bar{h}_{\boldsymbol{\theta}_h}^{\mathcal{C}_\epsilon}(\mathbf{x}), \\
{}^{\mathrm{CC}}\underline{h}_{\boldsymbol{\theta}_h}^{\mathcal{C}_\epsilon}(\alpha; \mathbf{x}) &:= (1 - \alpha) \, h_{\boldsymbol{\theta}_h}(\mathbf{x}_{\mathrm{adv}}) + \alpha \, \underline{h}_{\boldsymbol{\theta}_h}^{\mathcal{C}_\epsilon}(\mathbf{x}).
\end{aligned}
\tag{9}
$$

As we next show, $^{\mathrm{CC}}\mathcal{R}_{f_{\boldsymbol{\theta}}}^{\mathcal{C}_\epsilon}(\alpha; \mathbf{x}, y)$ can be realized by distilling onto $^{\mathrm{CC}}\bar{h}_{\boldsymbol{\theta}_h}^{\mathcal{C}_\epsilon}(\alpha; \mathbf{x})$ and $^{\mathrm{CC}}\underline{h}_{\boldsymbol{\theta}_h}^{\mathcal{C}_\epsilon}(\alpha; \mathbf{x})$.

**Proposition 3.2.** *The loss function*

$$
{}^{\mathrm{CC}}\mathcal{R}_{f_{\boldsymbol{\theta}}}^{\mathcal{C}_\epsilon}(\alpha; \mathbf{x}, y) := \sum_i \max \left\{ \left( {}^{\mathrm{CC}}\underline{h}_{\boldsymbol{\theta}_h}^{\mathcal{C}_\epsilon}(\alpha; \mathbf{x})_i - h_{\boldsymbol{\theta}_h^t}^t(\mathbf{x})_i \right)^2, \left( {}^{\mathrm{CC}}\bar{h}_{\boldsymbol{\theta}_h}^{\mathcal{C}_\epsilon}(\alpha; \mathbf{x})_i - h_{\boldsymbol{\theta}_h^t}^t(\mathbf{x})_i \right)^2 \right)
$$

*enjoys the following properties:*

1. $^{\mathrm{CC}}\mathcal{R}_{f_{\boldsymbol{\theta}}}^{\mathcal{C}_\epsilon}(0; \mathbf{x}, y) = \underline{\mathcal{R}}_{f_{\boldsymbol{\theta}}}^{\mathcal{C}_\epsilon}(\mathbf{x}, y)$ *and* $^{\mathrm{CC}}\mathcal{R}_{f_{\boldsymbol{\theta}}}^{\mathcal{C}_\epsilon}(1; \mathbf{x}, y) = \bar{\mathcal{R}}_{f_{\boldsymbol{\theta}}}^{\mathcal{C}_\epsilon}(\mathbf{x}, y)$;

2. $\underline{\mathcal{R}}_{f_{\boldsymbol{\theta}}}^{\mathcal{C}_\epsilon}(\mathbf{x}, y) \leq {}^{\mathrm{CC}}\mathcal{R}_{f_{\boldsymbol{\theta}}}^{\mathcal{C}_\epsilon}(\alpha; \mathbf{x}, y) \leq \bar{\mathcal{R}}_{f_{\boldsymbol{\theta}}}^{\mathcal{C}_\epsilon}(\mathbf{x}, y) \; \forall \, \alpha \in [0, 1]$;

3. $^{\mathrm{CC}}\mathcal{R}_{f_{\boldsymbol{\theta}}}^{\mathcal{C}_\epsilon}(\alpha; \mathbf{x}, y)$ *is continuous and monotonically increasing for* $\alpha \in [0, 1]$.

### 3.3 CC-DIST

While we expect special cases $^{\mathrm{CC}}\mathcal{R}_{f_{\boldsymbol{\theta}}}^{\mathcal{C}_\epsilon}(0; \mathbf{x}, y)$ and $^{\mathrm{CC}}\mathcal{R}_{f_{\boldsymbol{\theta}}}^{\mathcal{C}_\epsilon}(1; \mathbf{x}, y)$ to work well on settings where CC-IBP is employed with small or large $\alpha$ values, respectively, we aim to leverage the full flexibility from proposition 3.2 to obtain consistent performance across setups.

We will now show that, if the student uses an affine classification head, the CC-IBP convex combinations $^{\mathrm{CC}}\mathbf{z}_{f_{\boldsymbol{\theta}}}^{\mathcal{C}_\epsilon}(\alpha; \mathbf{x}, y)$ correspond to a simple affine function of $^{\mathrm{CC}}\underline{h}_{\boldsymbol{\theta}_h}^{\mathcal{C}_\epsilon}(\alpha; \mathbf{x})$ and $^{\mathrm{CC}}\bar{h}_{\boldsymbol{\theta}_h}^{\mathcal{C}_\epsilon}(\alpha; \mathbf{x})$, onto which distillation is performed. Consequently, we propose to employ the same $\alpha$ parameter for both $^{\mathrm{CC}}\mathcal{R}_{f_{\boldsymbol{\theta}}}^{\mathcal{C}_\epsilon}(\alpha; \mathbf{x}, y)$ and $^{\mathrm{CC}}\mathcal{L}_{f_{\boldsymbol{\theta}}}^{\mathcal{C}_\epsilon}(\alpha; \mathbf{x}, y)$, thus tightly coupling our distillation loss with CC-IBP.

**Lemma 3.3.** *Let the student classification head* $g_{\boldsymbol{\theta}_g}(\mathbf{a})$ *be affine, and let* $\tilde{g}_{\boldsymbol{\theta}_g}^y(\mathbf{a}) := \tilde{W}^n \mathbf{a} + \tilde{\mathbf{b}}^n$ *be its composition with the operator performing the difference between logits. We can write the CC-IBP convex combinations as:*

$$
{}^{\mathrm{CC}}\mathbf{z}_{f_{\boldsymbol{\theta}}}^{\mathcal{C}_\epsilon}(\alpha; \mathbf{x}, y) = \begin{bmatrix} [\tilde{W}^n]_+ & [\tilde{W}^n]_- \end{bmatrix} \begin{bmatrix} {}^{\mathrm{CC}}\underline{h}_{\boldsymbol{\theta}_h}^{\mathcal{C}_\epsilon}(\alpha; \mathbf{x}) \\ {}^{\mathrm{CC}}\bar{h}_{\boldsymbol{\theta}_h}^{\mathcal{C}_\epsilon}(\alpha; \mathbf{x}) \end{bmatrix} + \tilde{\mathbf{b}}^n.
$$

**CC-Dist loss**   Let $\beta$ be the distillation coefficient, determining the relative weight of the distillation term. Omitting any regularization, the overall training loss for CC-Dist (short for CC-IBP Distillation) takes the following form:

$$
\mathcal{L}^{\mathrm{CC\text{-}Dist}}(\alpha, \beta; \mathbf{x}, y) := {}^{\mathrm{CC}}\mathcal{L}_{f_{\boldsymbol{\theta}}}^{\mathcal{C}_\epsilon}(\alpha; \mathbf{x}, y) + \beta \; {}^{\mathrm{CC}}\mathcal{R}_{f_{\boldsymbol{\theta}}}^{\mathcal{C}_\epsilon}(\alpha; \mathbf{x}, y).
\tag{10}
$$

In practice, denoting by $w$ the dimensionality of the feature space, unless stated otherwise, we employ $\beta = {}^5/w$ in all reported experiments owing to its consistent performance across setups.

**Teacher models**   We propose to use teacher models trained via pure adversarial training (see §2.2) on the target task, employing the same architecture as the student. In order to comply with the conditions of lemma 3.3, we define the features as the activations before the last affine network layer.

**Intuition** We expect teachers to transfer part of their superior natural accuracy and empirical robustness to the target models. While the teacher certified robustness will be significantly smaller than for models trained via certified training, the CC-IBP term in equation (10) is employed to preserve student verifiability, hence increasing its certified robustness. Experimental evidence is provided in §5.2 and appendix E.1.

# 4 RELATED WORK

Owing to its favorable optimization properties (Jovanović et al., 2022), the relatively loose IBP (§2.1) is the method of choice (Mao et al., 2024a) when computing lower bounds to equation (2) to be employed for certified training. Tighter techniques are however preferred within branch-and-bound-based complete verifiers. Branch-and-bound (Bunel et al., 2018) couples a bounding algorithm with a strategy to refine the network relaxations by iteratively splitting problem (2) into subproblems (branching), which typically operates by splitting piecewise linearities into their linear components (De Palma et al., 2021; Henriksen & Lomuscio, 2021; Ferrari et al., 2022; Bunel et al., 2020b). Earlier bounding algorithms relied on linear network relaxations (Bunel et al., 2018; Ehlers, 2017; Anderson et al., 2020), with more effective techniques operating in the dual space (Dvijotham et al., 2018; Bunel et al., 2020a; De Palma et al., 2024a). State-of-the-art branch-and-bound frameworks have since converged to using fast bounds based on propagating couples of linear bounds through the network (Zhang et al., 2018; Singh et al., 2019; Xu et al., 2021) with Lagrangian relaxations employed to capture additional constraints (Wang et al., 2021; Ferrari et al., 2022; Zhang et al., 2022c; Zhou et al., 2024).

Earlier certified training works add geometric regularizers to the standard adversarial training loss (Xiao et al., 2019; Croce et al., 2019; Liu et al., 2021). Recent certified training schemes mix adversarial training with bounds from network relaxations to obtain strong post-training verifiability using branch-and-bound (see §2.2). Balunovic & Vechev (2020) perform adversarial attacks over latent-space over-approximations. The good performance of the latter at lower perturbation radii was matched through a regularizer on the area of network relaxations (De Palma et al., 2022), IBP-R, and later refined by connecting the gradients of the over-approximation with those of the attack (TAPS) (Mao et al., 2023). Another work, named SABR, proposed to compute network relaxations over a subset of $\mathcal{C}_\epsilon$ and to tune the subset size for each benchmark. This was then generalized into the notion of expressive losses (De Palma et al., 2024b) (see §3.1), which includes losses based on convex combinations forming the basis of this work. All the above methods train standard feedforward ReLU networks using convex relaxations, with IBP bounds being the most prevalent among them. Recent work has highlighted the superior representational power of networks trained via tighter relaxations (Baader et al., 2024; Mao et al., 2024b), and sought to overcome the associated optimization challenges (Balauca et al., 2025). Certifiably-robust networks can also be obtained by training 1-Lipschitz models using alternative architectures (Zhang et al., 2021; 2022a), the best-performing being SortNet (Zhang et al., 2022b). These were also shown to benefit from additional generated data (Altstidl et al., 2024), whose use is beyond the focus of our work. While we here concentrate on $\ell_\infty$ perturbations and deterministic certificates, 1-Lipschitz architectures (Prach et al., 2024; Meunier et al., 2022) and Lipschitz regularization (Hu et al., 2023; Leino et al.) are very effective when $\mathcal{C}_\epsilon$ is defined using the $\ell_2$ norm, setting under which probabilistic robustness certificates can be effectively obtained using randomised smoothing (Cohen et al., 2019).

Many specialized knowledge distillation schemes designed to transfer empirical robustness focus on modifying the logit-space distillation loss from equation (6). Assuming an adversarially-trained teacher, ARD (Goldblum et al., 2020) proposes to modify the KL term from equation (6) to evaluate the student on an adversarially-perturbed input. IAD (Zhu et al., 2022), instead, replaces the cross-entropy loss with a KL term between the student's perturbed and unperturbed outputs, weighted according to the teacher's inability to correctly classify the adversarial input. RSLAD (Zi et al., 2021) removes the dynamic weighting and trains with two KL terms between teacher and student: one using clean inputs, the other where the student is evaluated on a perturbation seeking to maximize the distance from the clean teacher output. In this context, less attention has been devoted to feature-space distillation methods (Muhammad et al., 2021). CRD (Vaishnavi et al., 2022) presented a logit-space distillation loss designed to transfer probabilistic certified robustness, relying on a sum of the student cross-entropy loss with an $\ell_2$ term on the models softmax outputs, all evaluated on random Gaussian perturbations. The focus of all the above works is to transfer some of the performance of larger

and more capable models onto smaller and less expensive architectures. However, it was shown that training a network for a given task, and then re-using it to distill knowledge onto a second model with the same architecture and training goal can improve performance through regularization. This was observed both in the context of standard training (Furlanello et al., 2018), and of pure adversarial training for empirical robustness (Chen et al., 2020). Finally, closely-related works at the intersection between knowledge distillation and deterministic certified robustness include: (i) investigating how robustness guarantees obtained on a student can be transferred from a distilled student to its teacher (Indri et al., 2024), and (ii) distilling a non-robust perceptual similarity metric onto a 1-Lipschitz architecture to obtain a certifiably-robust perceptual similarity metric (Ghazanfari et al., 2024). In this work, targeting supervised classification tasks from the robust vision literature and using techniques based on convex relaxations, we aim to leverage an empirically-robust yet hard-to-verify teacher to improve the deterministic certified robustness of a student with the same architecture.

Table 1: Evaluation of the effect of CC-Dist compared to pure CC-IBP (De Palma et al., 2024b) when training for certified robustness against $\ell_\infty$ norm perturbations. Literature results are provided as a reference. We highlight in bold the entries corresponding to the largest standard or certified accuracy for each benchmark, and, when they do not coincide, underline the best accuracies for ReLU-based architectures.

| Dataset | $\epsilon$ | Method | Source | Standard acc. [%] | Certified acc. [%] |
|---|---|---|---|---|---|
| CIFAR-10 | $\frac{2}{255}$ | CC-Dist | this work | **81.55** | **64.60** |
| | | CC-IBP | this work | 79.51 | 63.50 |
| | | CC-IBP | De Palma et al. (2024b) | 80.09 | 63.78 |
| | | MTL-IBP[†] | Mao et al. (2025) | 78.82 | 64.41 |
| | | STAPS | Mao et al. (2023) | 79.76 | 62.98 |
| | | SABR[†] | Mao et al. (2024a) | 79.89 | 63.28 |
| | | SortNet | Zhang et al. (2022b) | 67.72 | 56.94 |
| | | IBP-R[†] | Mao et al. (2024a) | 80.46 | 62.03 |
| | | IBP[†] | Mao et al. (2024a) | 68.06 | 56.18 |
| | | CROWN-IBP[†] | Mao et al. (2025) | 67.60 | 53.97 |
| | $\frac{8}{255}$ | CC-Dist | this work | **55.13** | 35.52 |
| | | CC-IBP | this work | 54.46 | 35.42 |
| | | CC-IBP | De Palma et al. (2024b) | 53.71 | 35.27 |
| | | MTL-IBP[†] | Mao et al. (2025) | 54.28 | 35.41 |
| | | STAPS | Mao et al. (2023) | 52.82 | 34.65 |
| | | SABR[†] | Mao et al. (2025) | 52.71 | 35.34 |
| | | SortNet | Zhang et al. (2022b) | 54.84 | **40.39** |
| | | IBP-R | De Palma et al. (2022) | 52.74 | 27.55 |
| | | IBP | Shi et al. (2021) | 48.94 | 34.97 |
| | | CROWN-IBP[†] | Mao et al. (2025) | 48.25 | 32.59 |
| TinyImageNet | $\frac{1}{255}$ | CC-Dist | this work | **43.78** | **27.88** |
| | | CC-IBP | this work | 41.28 | 26.53 |
| | | Exp-IBP | De Palma et al. (2024b) | 38.71 | 26.18 |
| | | MTL-IBP[†] | Mao et al. (2025) | 35.97 | 27.73 |
| | | STAPS[†] | Mao et al. (2025) | 30.63 | 22.31 |
| | | SABR[†] | De Palma et al. (2024b) | 38.68 | 25.85 |
| | | SortNet | Zhang et al. (2022b) | 25.69 | 18.18 |
| | | IBP[†] | Mao et al. (2025) | 26.77 | 19.82 |
| | | CROWN-IBP[†] | Mao et al. (2025) | 28.44 | 22.14 |
| ImageNet64 | $\frac{1}{255}$ | CC-Dist | this work | **28.17** | **13.96** |
| | | CC-IBP | this work | 25.94 | 13.69 |
| | | Exp-IBP | De Palma et al. (2024b) | 22.73 | 13.30 |
| | | SABR[†] | De Palma et al. (2024b) | 20.33 | 12.39 |
| | | SortNet | Zhang et al. (2022b) | 14.79 | 9.54 |
| | | CROWN-IBP | Xu et al. (2020) | 16.23 | 8.73 |
| | | IBP | Gowal et al. (2019) | 15.96 | 6.13 |

[†] Evaluation from later work attaining a larger standard or certified accuracy than reported in the original work.

## 5 EXPERIMENTAL EVALUATION

This section presents an experimental evaluation of the proposed distillation scheme, focusing on image classification datasets from the certified training literature. We first present results on the effectiveness of our novel distillation loss (§5.1), followed by a comparison of the trained models with their teachers (§5.2), and by an analysis of the effect of the distillation coefficient on CC-Dist (§5.3).

In line with previous work (Müller et al., 2023; Mao et al., 2023; Shi et al., 2021; De Palma et al., 2024b), all the models trained via CC-Dist in this evaluation are based a 7-layer convolutional architecture named `CNN-7`, regularized using an $\ell_1$ term. The teacher models are trained via adversarial training based on a 10-step PGD adversary (Madry et al., 2018) and $\ell_1$ regularization. As this was found to be beneficial in practice, we employ the `CNN-7` architecture for the teacher model too. We implemented CC-Dist in PyTorch (Paszke et al., 2019) similarly to the expressive losses codebase (De Palma et al., 2024b), also relying on `auto_LiRPA` (Xu et al., 2020) to compute IBP bounds. As common in the wider certified training literature (Müller et al., 2023; Mao et al., 2023), we employ the custom regularization and initialization introduced by Shi et al. (2021). Post-training verification is performed using the OVAL branch-and-bound framework (Bunel et al., 2018; 2020a; De Palma et al., 2021) similarly to De Palma et al. (2022; 2024b), using a configuration based on $\alpha$-$\beta$-CROWN network bounds (Wang et al., 2021; Xu et al., 2021) and employing at most 600 seconds per image. Additional details and supplementary experiments can be found in appendices D and E, respectively.

### 5.1 CC-DIST EVALUATION

We now evaluate the performance of CC-Dist (§3.3) with respect to pure CC-IBP (De Palma et al., 2024b), and compare the resulting performance with results taken from the certified training literature. For each method from the literature, we report the best attained performance across previous works and network architectures. In order to isolate the effect of any modification within our codebase and experimental setup, we report both CC-IBP results from our evaluations, and the literature results corresponding to the best-performing expressive loss (CC-IBP, MTL-IBP, or Exp-IBP) from the original work (De Palma et al., 2024b). Table 1 shows that the distillation loss improves on both the standard and the certified accuracies of pure CC-IBP across all the considered CIFAR-10 (Krizhevsky & Hinton, 2009), TinyImageNet (Le & Yang, 2015), and downscaled ImageNet ($64 \times 64$) (Chrabaszcz et al., 2017) benchmarks. Consequently, CC-Dist establishes a new state-of-the-art, improving on both standard and certified accuracy compared to previously-reported results, on all benchmarks except CIFAR-10 with $\epsilon = 8/255$. Differently from the other settings, the best certified accuracy on this benchmark is attained by SortNet (Zhang et al., 2022b), which uses a specialized 1-Lipschitz network architecture (see §4). Its performance on this setup can be further improved through the use of additional generated data (Altstidl et al., 2024), which is outside the scope of our work. While the sub-par performance of ReLU networks on this benchmark is well known in the literature (Müller et al., 2023; De Palma et al., 2024b), we are hopeful that distillation from empirically-robust teachers may be beneficial to 1-Lipschitz networks too, and defer the investigation to future work. On TinyImageNet and downscaled ImageNet we found that both the standard and certified accuracies of CC-IBP benefit from a smaller $\alpha$ coefficient than those employed in De Palma et al. (2024b), explaining the difference between our CC-IBP results and those reported in the original work. Owing to the combined effect of this and of the benefits of our novel distillation loss, the difference between CC-Dist and the results from the literature is particularly remarkable on these two larger-scale benchmarks.

Appendix E.1 sheds light on the success of our distillation technique by analyzing differences in empirical robustness and IBP regularization between CC-IBP and CC-Dist. Appendices E.2, E.3 and E.8 demonstrate the wider potential of knowledge distillation for certified training by showing (i) the effectiveness of special cases of the proposed distillation loss, (ii) that SABR, MTL-IBP, and IBP-R benefit from a similar distillation process and (iii) that CC-Dist is effective on a different model architecture and activation function. Appendices E.4 and E.9 respectively investigate the performance of logit-based distillation, and of using earlier features for ${}^{\text{CC}}\mathcal{R}_{f_{\boldsymbol{\theta}}}^{\mathcal{C}_\epsilon}(\alpha; \mathbf{x}, y)$. Appendix E.10 shows that employing clean teachers results in worse performance profiles, highlighting the importance of robust representations.

Table 2: Comparison between the CC-Dist student models from table 1 and the respective teacher models.

| Dataset | $\epsilon$ | Method | Standard acc. [%] | AA acc. [%] | Certified acc.[†] [%] |
|---|---|---|---|---|---|
| CIFAR-10 | $\frac{2}{255}$ | STUDENT | 81.55 | 70.71 | 64.4 |
| | | TEACHER | 88.23 | 73.26 | 1.2 |
| | $\frac{8}{255}$ | STUDENT | 55.13 | 36.80 | 35.8 |
| | | TEACHER | 78.16 | 41.62 | 0.0 |
| TinyImageNet | $\frac{1}{255}$ | STUDENT | 43.78 | 32.30 | 32.6 |
| | | TEACHER | 47.18 | 34.17 | 17.4 |
| ImageNet64 | $\frac{1}{255}$ | STUDENT | 28.17 | 17.87 | 5.6 |
| | | TEACHER | 40.30 | 25.34 | 0.0 |

[†] Certified accuracy reported over the first 500 test images.

## 5.2 TEACHER-STUDENT COMPARISON

In order to provide insights on the distillation process, we now present a comparison of the performance of the CC-Dist models from table 1 with that of the respective teacher models. Owing to the extremely large computational cost associated to running branch-and-bound on networks trained via pure adversarial training (the teachers), we restrict the certified robustness comparison on the first 500 images of the test set of each benchmark. Table 2 shows a clear difference in terms of standard accuracy and empirical robustness to AutoAttack (Croce & Hein, 2020) between teacher and student, the latter displaying worse performance as expected. On the other hand, the teachers fail to attain good certified robustness across any of the considered benchmarks, highlighting that the students significantly surpass the respective teachers in terms of verifiability. In other words, the distillation process successfully exploits empirically-robust teachers to improve the state-of-the-art in certifiably-robust models.

## 5.3 SENSITIVITY TO DISTILLATION COEFFICIENT

We now present an empirical study of the effect of the distillation coefficient $\beta$ on the performance profiles of CC-Dist. As mentioned in §3.3, we keep $\beta = 5/w$ throughout all the experiments, where $w$ denotes the dimensionality of the feature space. This is in order to avoid the introduction of an additional hyper-parameter, and owing to its strong performance across the considered settings. Figure 1 compares the behavior at $\beta = 5/w$ from table 1, where both certified robustness via branch-and-bound and standard performance are enhanced, with results for different distillation coefficients. It reports standard performance, empirical adversarial robustness (measured through the attacks within the employed branch-and-bound framework), and certified robustness (both using branch-and-bound and the best results between incomplete verifiers CROWN (Zhang et al., 2018) and IBP (Gowal et al., 2019)) across $\beta$ values on the CIFAR-10 test set. For $\epsilon = 2/255$, increasing $\beta$ leads to a steady increase in both standard accuracy and empirical robustness. At the same time, verifiability increases until an intermediate $\beta$ value, then steadily decreases, with the certified accuracy via incomplete verifiers peaking at a lower value than when using branch-and-bound. This

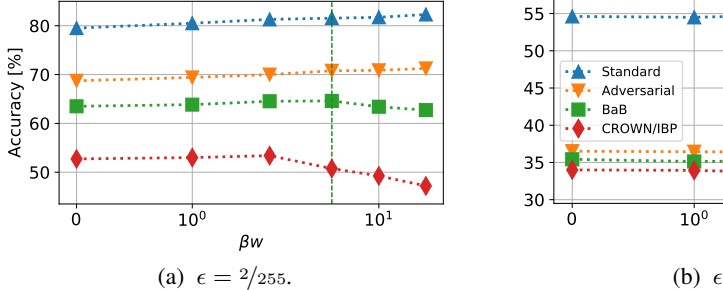

(a) $\epsilon = 2/255$.

(b) $\epsilon = 8/255$.

Figure 1: Standard, empirical adversarial and certified accuracies (BaB and CROWN/IBP) under $\ell_\infty$ perturbations of networks trained using CC-Dist under varying distillation coefficient $\beta$. The legend is reported once for all subfigures in plot 1(b). Metrics are reported on the CIFAR-10 test set. The $\beta$ value employed throughout the paper (see §3.3) is marked by a dashed vertical line.

is similar to the behavior of the $\alpha$ parameter in expressive losses such as CC-IBP (De Palma et al., 2024b). As also visible in table 1, distillation appears to be less helpful for $\epsilon = 8/255$, where standard performance is enhanced only for intermediate $\beta$ values. We explain this difference through the larger degree of similarity between teacher and student on $\epsilon = 2/255$: see table 2.

## 6 CONCLUSIONS

We presented a novel training scheme, named CC-Dist, that successfully leverages empirically-robust models to train better certifiably-robust neural networks through knowledge distillation. Complementing CC-IBP, CC-Dist relies on a versatile feature-space distillation loss that can operate on convex combinations between bounds from student convex relaxations and adversarial student features. We show that CC-Dist improves on both the standard and certified robust accuracies of CC-IBP on all considered benchmarks from the robust vision literature, in spite of the relatively low certified robustness of the employed teacher models. As a result, CC-Dist attains a new state-of-the-art in certified training across ReLU architectures, showcasing the potential of knowledge distillation for certified robustness. While our work focuses on teachers that employ the same architecture as the student, we are hopeful that further progress can be made by appropriately training teachers with larger effective capacity.

## ETHICS STATEMENT

We do not anticipate any short-term negative impact of certifiably-robust networks. Indeed, we believe that efficient certified training techniques should be primarily leveraged towards providing guarantees to deep learning systems operating in safety-critical contexts. Nevertheless, we acknowledge that adversarial attacks may have social utility when deployed against unethical systems, which constitute unintended use of machine learning technologies, pointing to a potential shortcoming of provable robustness. Effective mitigation strategies may include targeted regulations.

## REPRODUCIBILITY STATEMENT

Code for CC-Dist is provided as part of the supplementary material, and pseudo-code is provided in appendix C. Information to reproduce the experiments can be found in in §5 and appendix D, which also includes details on the employed compute resources and software acknowledgments. Owing to the large cost associated with the verification experiments (a timeout of $600$ seconds per evaluation image is employed), and complying with related previous works (De Palma et al., 2024b; Mao et al., 2023; Müller et al., 2023; Mao et al., 2024a; 2025), we provide single-seed results. An indication of experimental variability on a single dataset is nevertheless provided in appendix E.6.

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

## A   INTERVAL BOUND PROPAGATION

We here provide details concerning the computations of IBP bounds omitted from §2.1. Assuming a feed-forward network structure, let $W^j$ and $b^j$ respectively be the weight and bias of the $j$-th layer, and let $\sigma$ denote a monotonic element-wise activation function, which, as stated in §2, we assume to be the ReLU activation in this work. Furthermore, let $\tilde{W}^n := \left( {}_y I^k - I^k \right) W^n$ and $\tilde{\mathbf{b}}^n := \left( {}_y I^k - I^k \right) \mathbf{b}^n$ denote the weight and bias corresponding to the composition of the last network layer and of the difference between logits. IBP proceeds by first computing $\hat{\mathbf{l}}^1 = W^1 \mathbf{x} - \epsilon \left| W^1 \right| \mathbf{1} + \mathbf{b}^1$ and $\hat{\mathbf{u}}^1 = W^1 \mathbf{x} + \epsilon \left| W^1 \right| \mathbf{1} + \mathbf{b}^1$, and then derives $\underline{\mathbf{z}}_{f_{\boldsymbol{\theta}}}^{\mathcal{C}_\epsilon}(\mathbf{x}, y)$ by iteratively computing lower and upper bounds ($\hat{\mathbf{l}}^j$, and $\hat{\mathbf{u}}^j$, respectively) to the network pre-activations at layer $j$:

$$\underline{\mathbf{z}}_{f_{\boldsymbol{\theta}}}^{\mathcal{C}_\epsilon}(\mathbf{x}, y) = [\tilde{W}^n]_+ \, \sigma\left( \hat{\mathbf{l}}^{n-1} \right) + [\tilde{W}^n]_- \, \sigma\left( \hat{\mathbf{u}}^{n-1} \right) + \tilde{\mathbf{b}}^n, \quad \text{where:}$$

$$\left[ \begin{array}{l} \hat{\mathbf{l}}^j = [W^j]_+ \, \sigma\left( \hat{\mathbf{l}}^{j-1} \right) + \frac{1}{2}[W^j]_- \, \sigma\left( \hat{\mathbf{u}}^{j-1} \right) + \mathbf{b}_j \\ \hat{\mathbf{u}}^j = [W^j]_+ \, \sigma\left( \hat{\mathbf{u}}^{j-1} \right) + \frac{1}{2}[W^j]_- \, \sigma\left( \hat{\mathbf{l}}^{j-1} \right) + \mathbf{b}_j \end{array} \right] \forall \, j \in [\![ 2, n-1 ]\!] . \tag{11}$$

## B   PROOFS OF TECHNICAL RESULTS

We here provide the proofs omitted from §3.

*Remark* B.1. Let $f : \mathbb{R}^n \to \mathbb{R}$. If $\mathcal{A} \neq \emptyset$ and $\mathcal{A} \subseteq \mathcal{B}$, then $\max_{\mathbf{x} \in \mathcal{A}} f(\mathbf{x}) \leq \max_{\mathbf{x} \in \mathcal{B}} f(\mathbf{x})$.

**Proposition 3.1.** *Let $\underline{h}_{\boldsymbol{\theta}_h}^{\mathcal{C}_\epsilon}(\mathbf{x})$ and $\bar{h}_{\boldsymbol{\theta}_h}^{\mathcal{C}_\epsilon}(\mathbf{x})$ respectively denote IBP lower and upper bounds to the student features:*

$$\underline{h}_{\boldsymbol{\theta}_h}^{\mathcal{C}_\epsilon}(\mathbf{x}) \leq h_{\boldsymbol{\theta}_h}(\mathbf{x}') \leq \bar{h}_{\boldsymbol{\theta}_h}^{\mathcal{C}_\epsilon}(\mathbf{x}), \quad \forall \mathbf{x}' \in \mathcal{C}_\epsilon(\mathbf{x}).$$

*The loss function $\bar{\mathcal{R}}_{f_{\boldsymbol{\theta}}}^{\mathcal{C}_\epsilon}(\mathbf{x}, y) := \sum_i \max \left\{ \left( \underline{h}_{\boldsymbol{\theta}_h}^{\mathcal{C}_\epsilon}(\mathbf{x})_i - h_{\boldsymbol{\theta}_h^t}^t(\mathbf{x})_i \right)^2, \left( \bar{h}_{\boldsymbol{\theta}_h}^{\mathcal{C}_\epsilon}(\mathbf{x})_i - h_{\boldsymbol{\theta}_h^t}^t(\mathbf{x})_i \right)^2 \right\}$ is an upper bound to the worst-case distillation loss from equation (8): $\bar{\mathcal{R}}_{f_{\boldsymbol{\theta}}}^{\mathcal{C}_\epsilon}(\mathbf{x}, y) \geq \mathcal{R}_{f_{\boldsymbol{\theta}}}^{\mathcal{C}_\epsilon}(\mathbf{x}, y)$.*

*Proof.* Let us start by upper bounding the worst-case distillation loss $\mathcal{R}_{f_{\boldsymbol{\theta}}}^{\mathcal{C}_\epsilon}(\mathbf{x}, y)$:

$$\mathcal{R}_{f_{\boldsymbol{\theta}}}^{\mathcal{C}_\epsilon}(\mathbf{x}, y) = \max_{\mathbf{x}' \in \mathcal{C}_\epsilon(\mathbf{x})} \left\| h_{\boldsymbol{\theta}_h}(\mathbf{x}') - h_{\boldsymbol{\theta}_h^t}^t(\mathbf{x}) \right\|_2^2 = \max_{\mathbf{x}' \in \mathcal{C}_\epsilon(\mathbf{x})} \left[ \sum_i \left( h_{\boldsymbol{\theta}_h}(\mathbf{x}')_i - h_{\boldsymbol{\theta}_h^t}^t(\mathbf{x})_i \right)^2 \right]$$

$$\leq \sum_i \max_{\mathbf{x}' \in \mathcal{C}_\epsilon(\mathbf{x})} \left( h_{\boldsymbol{\theta}_h}(\mathbf{x}')_i - h_{\boldsymbol{\theta}_h^t}^t(\mathbf{x})_i \right)^2 .$$

Let $\mathcal{M}_i = \left\{ h_{\boldsymbol{\theta}_h}(\mathbf{x}')_i \,\middle|\, \mathbf{x}' \in \arg\max_{\mathbf{x}' \in \mathcal{C}_\epsilon(\mathbf{x})} \left( h_{\boldsymbol{\theta}_h}(\mathbf{x}')_i - h_{\boldsymbol{\theta}_h^t}^t(\mathbf{x})_i \right)^2 \right\}$. By the definition of the IBP feature bounds $\underline{h}_{\boldsymbol{\theta}_h}^{\mathcal{C}_\epsilon}(\mathbf{x})$ and $\bar{h}_{\boldsymbol{\theta}_h}^{\mathcal{C}_\epsilon}(\mathbf{x})$:

$$\mathcal{M}_i \subseteq \left[ \underline{h}_{\boldsymbol{\theta}_h}^{\mathcal{C}_\epsilon}(\mathbf{x})_i, \bar{h}_{\boldsymbol{\theta}_h}^{\mathcal{C}_\epsilon}(\mathbf{x})_i \right] .$$

Recalling remark B.1, we can hence further bound $\mathcal{R}_{f_{\boldsymbol{\theta}}}^{\mathcal{C}_\epsilon}(\mathbf{x}, y)$ as follows:

$$\mathcal{R}_{f_{\boldsymbol{\theta}}}^{\mathcal{C}_\epsilon}(\mathbf{x}, y) \leq \sum_i \max_{\mathbf{x}' \in \mathcal{C}_\epsilon(\mathbf{x})} \left( h_{\boldsymbol{\theta}_h}(\mathbf{x}')_i - h_{\boldsymbol{\theta}_h^t}^t(\mathbf{x})_i \right)^2 = \sum_i \max_{h \in \mathcal{M}_i} \left( h - h_{\boldsymbol{\theta}_h^t}^t(\mathbf{x})_i \right)^2$$

$$\leq \sum_i \max_{h \in \left[ \underline{h}_{\boldsymbol{\theta}_h}^{\mathcal{C}_\epsilon}(\mathbf{x})_i, \bar{h}_{\boldsymbol{\theta}_h}^{\mathcal{C}_\epsilon}(\mathbf{x})_i \right]} \left( h - h_{\boldsymbol{\theta}_h^t}^t(\mathbf{x})_i \right)^2 .$$

Let us note that (for any $b, l, u \in \mathbb{R}, l \leq u$):

$$\max_{x \in [l, u]} (x - b)^2 = \max \left\{ (l - b)^2, (u - b)^2 \right\} . \tag{12}$$

We can hence write $\mathcal{R}_{f_{\boldsymbol{\theta}}}^{\mathcal{C}_\epsilon}(\mathbf{x}, y) \leq \bar{\mathcal{R}}_{f_{\boldsymbol{\theta}}}^{\mathcal{C}_\epsilon}(\mathbf{x}, y)$. $\square$

**Proposition 3.2.** *The loss function*

$$^{CC}\mathcal{R}_{f_{\boldsymbol{\theta}}}^{\mathcal{C}_\epsilon}(\alpha; \mathbf{x}, y) := \sum_i \max\left\{ \left( {}^{CC}\underline{h}_{\boldsymbol{\theta}_h}^{\mathcal{C}_\epsilon}(\alpha; \mathbf{x})_i - h_{\boldsymbol{\theta}_h^t}^t(\mathbf{x})_i \right)^2, \left( {}^{CC}\bar{h}_{\boldsymbol{\theta}_h}^{\mathcal{C}_\epsilon}(\alpha; \mathbf{x})_i - h_{\boldsymbol{\theta}_h^t}^t(\mathbf{x})_i \right)^2 \right)$$

*enjoys the following properties:*

1. $^{CC}\mathcal{R}_{f_{\boldsymbol{\theta}}}^{\mathcal{C}_\epsilon}(0; \mathbf{x}, y) = \underline{\mathcal{R}}_{f_{\boldsymbol{\theta}}}^{\mathcal{C}_\epsilon}(\mathbf{x}, y)$ *and* $^{CC}\mathcal{R}_{f_{\boldsymbol{\theta}}}^{\mathcal{C}_\epsilon}(1; \mathbf{x}, y) = \bar{\mathcal{R}}_{f_{\boldsymbol{\theta}}}^{\mathcal{C}_\epsilon}(\mathbf{x}, y);$

2. $\underline{\mathcal{R}}_{f_{\boldsymbol{\theta}}}^{\mathcal{C}_\epsilon}(\mathbf{x}, y) \leq {}^{CC}\mathcal{R}_{f_{\boldsymbol{\theta}}}^{\mathcal{C}_\epsilon}(\alpha; \mathbf{x}, y) \leq \bar{\mathcal{R}}_{f_{\boldsymbol{\theta}}}^{\mathcal{C}_\epsilon}(\mathbf{x}, y) \; \forall \, \alpha \in [0, 1];$

3. $^{CC}\mathcal{R}_{f_{\boldsymbol{\theta}}}^{\mathcal{C}_\epsilon}(\alpha; \mathbf{x}, y)$ *is continuous and monotonically increasing for* $\alpha \in [0, 1]$.

*Proof.* Let us define $\mathcal{I}_i^\alpha := \left[ {}^{CC}\underline{h}_{\boldsymbol{\theta}_h}^{\mathcal{C}_\epsilon}(\alpha; \mathbf{x})_i, \; {}^{CC}\bar{h}_{\boldsymbol{\theta}_h}^{\mathcal{C}_\epsilon}(\alpha; \mathbf{x})_i \right]$. Owing to equation (12), we can write the following alternative definition of $^{CC}\mathcal{R}_{f_{\boldsymbol{\theta}}}^{\mathcal{C}_\epsilon}(\alpha; \mathbf{x}, y)$:

$$^{CC}\mathcal{R}_{f_{\boldsymbol{\theta}}}^{\mathcal{C}_\epsilon}(\alpha; \mathbf{x}, y) = \sum_i \max_{h \in \mathcal{I}_i^\alpha} \left( h - h_{\boldsymbol{\theta}_h^t}^t(\mathbf{x})_i \right)^2. \tag{13}$$

Let us recall that $^{CC}\bar{h}_{\boldsymbol{\theta}_h}^{\mathcal{C}_\epsilon}(\alpha; \mathbf{x}) := (1 - \alpha) \, h_{\boldsymbol{\theta}_h}(\mathbf{x}_{\text{adv}}) + \alpha \, \bar{h}_{\boldsymbol{\theta}_h}^{\mathcal{C}_\epsilon}(\mathbf{x})$, and $^{CC}\underline{h}_{\boldsymbol{\theta}_h}^{\mathcal{C}_\epsilon}(\alpha; \mathbf{x}) := (1 - \alpha) \, h_{\boldsymbol{\theta}_h}(\mathbf{x}_{\text{adv}}) + \alpha \, \underline{h}_{\boldsymbol{\theta}_h}^{\mathcal{C}_\epsilon}(\mathbf{x})$. By pointing out that $h_{\boldsymbol{\theta}_h}(\mathbf{x}_{\text{adv}}) \in \left[ \underline{h}_{\boldsymbol{\theta}_h}^{\mathcal{C}_\epsilon}(\mathbf{x}), \bar{h}_{\boldsymbol{\theta}_h}^{\mathcal{C}_\epsilon}(\mathbf{x}) \right]$, we can see that, $\forall \alpha \in [0, 1]$:

$$\underline{h}_{\boldsymbol{\theta}_h}^{\mathcal{C}_\epsilon}(\mathbf{x}) \leq {}^{CC}\underline{h}_{\boldsymbol{\theta}_h}^{\mathcal{C}_\epsilon}(\alpha; \mathbf{x}) \leq h_{\boldsymbol{\theta}_h}(\mathbf{x}_{\text{adv}}), \quad h_{\boldsymbol{\theta}_h}(\mathbf{x}_{\text{adv}}) \leq {}^{CC}\bar{h}_{\boldsymbol{\theta}_h}^{\mathcal{C}_\epsilon}(\alpha; \mathbf{x}) \leq \bar{h}_{\boldsymbol{\theta}_h}^{\mathcal{C}_\epsilon}(\mathbf{x}).$$

Consequently, $\forall \alpha \in [0, 1]$ and $\forall i \in [\![1, m]\!]$ ($m$ being the dimensionality of the feature space):

$$\mathcal{I}_i^0 = \{h_{\boldsymbol{\theta}_h}(\mathbf{x}_{\text{adv}})_i\} \subseteq \mathcal{I}_i^\alpha \subseteq \left[ \underline{h}_{\boldsymbol{\theta}_h}^{\mathcal{C}_\epsilon}(\mathbf{x})_i, \bar{h}_{\boldsymbol{\theta}_h}^{\mathcal{C}_\epsilon}(\mathbf{x})_i \right] = \mathcal{I}_i^1.$$

And hence, as per remark B.1:

$$\sum_i \max_{h \in \mathcal{I}_i^0} \left( h - h_{\boldsymbol{\theta}_h^t}^t(\mathbf{x})_i \right)^2 \leq \sum_i \max_{h \in \mathcal{I}_i^\alpha} \left( h - h_{\boldsymbol{\theta}_h^t}^t(\mathbf{x})_i \right)^2 \leq \sum_i \max_{h \in \mathcal{I}_i^1} \left( h - h_{\boldsymbol{\theta}_h^t}^t(\mathbf{x})_i \right)^2. \tag{14}$$

Using equation (13) with $\alpha = 0$ we can see that:

$$^{CC}\mathcal{R}_{f_{\boldsymbol{\theta}}}^{\mathcal{C}_\epsilon}(0; \mathbf{x}, y) = \sum_i \max_{h \in \mathcal{I}_i^0} \left( h - h_{\boldsymbol{\theta}_h^t}^t(\mathbf{x})_i \right)^2 = \sum_i \left( h_{\boldsymbol{\theta}_h}(\mathbf{x}_{\text{adv}})_i - h_{\boldsymbol{\theta}_h^t}^t(\mathbf{x})_i \right)^2$$

$$= \left\| h_{\boldsymbol{\theta}_h}(\mathbf{x}_{\text{adv}}) - h_{\boldsymbol{\theta}_h^t}^t(\mathbf{x}) \right\|_2^2 = \underline{\mathcal{R}}_{f_{\boldsymbol{\theta}}}^{\mathcal{C}_\epsilon}(\mathbf{x}, y).$$

Using it again with $\alpha = 1$ and recalling equation (12), we obtain:

$$^{CC}\mathcal{R}_{f_{\boldsymbol{\theta}}}^{\mathcal{C}_\epsilon}(1; \mathbf{x}, y) = \sum_i \max_{h \in \mathcal{I}_i^1} \left( h - h_{\boldsymbol{\theta}_h^t}^t(\mathbf{x})_i \right)^2 =$$

$$= \sum_i \max\left\{ \left( \underline{h}_{\boldsymbol{\theta}_h}^{\mathcal{C}_\epsilon}(\mathbf{x})_i - h_{\boldsymbol{\theta}_h^t}^t(\mathbf{x})_i \right)^2, \left( \bar{h}_{\boldsymbol{\theta}_h}^{\mathcal{C}_\epsilon}(\mathbf{x})_i - h_{\boldsymbol{\theta}_h^t}^t(\mathbf{x})_i \right)^2 \right\} = \bar{\mathcal{R}}_{f_{\boldsymbol{\theta}}}^{\mathcal{C}_\epsilon}(\mathbf{x}, y),$$

Hence proving the first point of the proposition.

The second point can be now proved by again using equation (13) within equation (14):

$$\underline{\mathcal{R}}_{f_{\boldsymbol{\theta}}}^{\mathcal{C}_\epsilon}(\mathbf{x}, y) = {}^{CC}\mathcal{R}_{f_{\boldsymbol{\theta}}}^{\mathcal{C}_\epsilon}(0; \mathbf{x}, y) \leq {}^{CC}\mathcal{R}_{f_{\boldsymbol{\theta}}}^{\mathcal{C}_\epsilon}(\alpha; \mathbf{x}, y) \leq {}^{CC}\mathcal{R}_{f_{\boldsymbol{\theta}}}^{\mathcal{C}_\epsilon}(1; \mathbf{x}, y) = \bar{\mathcal{R}}_{f_{\boldsymbol{\theta}}}^{\mathcal{C}_\epsilon}(\mathbf{x}, y).$$

Concerning the third point, the continuity of $^{CC}\mathcal{R}_{f_{\boldsymbol{\theta}}}^{\mathcal{C}_\epsilon}(\alpha; \mathbf{x}, y)$ for $\alpha \in [0, 1]$ can be proved by pointing out that $^{CC}\mathcal{R}_{f_{\boldsymbol{\theta}}}^{\mathcal{C}_\epsilon}(\alpha; \mathbf{x}, y)$ is composed of operators preserving continuity (pointwise $\max$), linear combinations and compositions of continuous functions of $\alpha$. In order to prove monotonicity, note that,

for any $\alpha, \alpha' \in [0, 1]$ with $\alpha \leq \alpha'$, ${}^{\mathrm{CC}}\bar{h}_{\boldsymbol{\theta}_h}^{\mathcal{C}_\epsilon}(\alpha'; \mathbf{x}) \geq {}^{\mathrm{CC}}\bar{h}_{\boldsymbol{\theta}_h}^{\mathcal{C}_\epsilon}(\alpha; \mathbf{x})$ and ${}^{\mathrm{CC}}\underline{h}_{\boldsymbol{\theta}_h}^{\mathcal{C}_\epsilon}(\alpha'; \mathbf{x}) \leq {}^{\mathrm{CC}}\underline{h}_{\boldsymbol{\theta}_h}^{\mathcal{C}_\epsilon}(\alpha; \mathbf{x})$, implying $\mathcal{I}_i^\alpha \subseteq \mathcal{I}_i^{\alpha'}$. Hence, recalling remark B.1, for any $\alpha, \alpha' \in [0, 1]$ with $\alpha \leq \alpha'$, we have:

$$
{}^{\mathrm{CC}}\mathcal{R}_{f_{\boldsymbol{\theta}}}^{\mathcal{C}_\epsilon}(\alpha; \mathbf{x}, y) = \sum_i \max_{h \in \mathcal{I}_i^\alpha} \left( h - h_{\boldsymbol{\theta}_h^t}^t(\mathbf{x})_i \right)^2 \leq \sum_i \max_{h \in \mathcal{I}_i^{\alpha'}} \left( h - h_{\boldsymbol{\theta}_h^t}^t(\mathbf{x})_i \right)^2 = {}^{\mathrm{CC}}\mathcal{R}_{f_{\boldsymbol{\theta}}}^{\mathcal{C}_\epsilon}(\alpha'; \mathbf{x}, y),
$$

which proves that ${}^{\mathrm{CC}}\mathcal{R}_{f_{\boldsymbol{\theta}}}^{\mathcal{C}_\epsilon}(\alpha; \mathbf{x}, y)$ is monotonically increasing for $\alpha \in [0, 1]$. $\qquad\square$

**Lemma 3.3.** *Let the student classification head $g_{\boldsymbol{\theta}_g}(\mathbf{a})$ be affine, and let $\tilde{g}_{\boldsymbol{\theta}_g}^y(\mathbf{a}) := \tilde{W}^n \mathbf{a} + \tilde{\mathbf{b}}^n$ be its composition with the operator performing the difference between logits. We can write the CC-IBP convex combinations as:*

$$
{}^{\mathrm{CC}}\mathbf{z}_{f_{\boldsymbol{\theta}}}^{\mathcal{C}_\epsilon}(\alpha; \mathbf{x}, y) = \left[ \begin{array}{cc} [\tilde{W}^n]_+ & [\tilde{W}^n]_- \end{array} \right] \left[ \begin{array}{c} {}^{\mathrm{CC}}\underline{h}_{\boldsymbol{\theta}_h}^{\mathcal{C}_\epsilon}(\alpha; \mathbf{x}) \\ {}^{\mathrm{CC}}\bar{h}_{\boldsymbol{\theta}_h}^{\mathcal{C}_\epsilon}(\alpha; \mathbf{x}) \end{array} \right] + \tilde{\mathbf{b}}^n.
$$

*Proof.* We can write the student logit differences as: $\mathbf{z}_{f_{\boldsymbol{\theta}}}(\mathbf{x}, y) = \tilde{W}^n h_{\boldsymbol{\theta}_h}(\mathbf{x}) + \tilde{\mathbf{b}}^n$. Hence, given IBP bounds to the student features $\underline{h}_{\boldsymbol{\theta}_h}^{\mathcal{C}_\epsilon}(\mathbf{x})$ and $\bar{h}_{\boldsymbol{\theta}_h}^{\mathcal{C}_\epsilon}(\mathbf{x})$ (corresponding to $\sigma\left(\hat{\mathbf{l}}^{n-1}\right)$ and $\sigma\left(\hat{\mathbf{u}}^{n-1}\right)$ in equation (11), respectively), we can compute the logit differences lower bounds $\underline{\mathbf{z}}_{f_{\boldsymbol{\theta}}}^{\mathcal{C}_\epsilon}(\mathbf{x}, y)$ as per equation (11):

$$
\underline{\mathbf{z}}_{f_{\boldsymbol{\theta}}}^{\mathcal{C}_\epsilon}(\mathbf{x}, y) = [\tilde{W}^n]_+ \underline{h}_{\boldsymbol{\theta}_h}^{\mathcal{C}_\epsilon}(\mathbf{x}) + [\tilde{W}^n]_- \bar{h}_{\boldsymbol{\theta}_h}^{\mathcal{C}_\epsilon}(\mathbf{x}) + \tilde{\mathbf{b}}^n.
$$

Let us write the adversarial logit differences $\mathbf{z}_{f_{\boldsymbol{\theta}}}(\mathbf{x}_{\mathrm{adv}}, y)$ as a function of the adversarial student features:

$$
\mathbf{z}_{f_{\boldsymbol{\theta}}}(\mathbf{x}_{\mathrm{adv}}, y) = \tilde{W}^n h_{\boldsymbol{\theta}_h}(\mathbf{x}_{\mathrm{adv}}) + \tilde{\mathbf{b}}^n = \left( [\tilde{W}^n]_+ + [\tilde{W}^n]_- \right) h_{\boldsymbol{\theta}_h}(\mathbf{x}_{\mathrm{adv}}) + \tilde{\mathbf{b}}^n.
$$

Replacing the above two equations in the definition of the CC-IBP convex combinations ${}^{\mathrm{CC}}\mathbf{z}_{f_{\boldsymbol{\theta}}}^{\mathcal{C}_\epsilon}(\alpha; \mathbf{x}, y)$ (see §3.1) we get:

$$
\begin{aligned}
{}^{\mathrm{CC}}\mathbf{z}_{f_{\boldsymbol{\theta}}}^{\mathcal{C}_\epsilon}(\alpha; \mathbf{x}, y) &= (1 - \alpha)\, \mathbf{z}_{f_{\boldsymbol{\theta}}}(\mathbf{x}_{\mathrm{adv}}, y) + \alpha\, \underline{\mathbf{z}}_{f_{\boldsymbol{\theta}}}^{\mathcal{C}_\epsilon}(\mathbf{x}, y) \\
&= (1 - \alpha) \left[ \left( [\tilde{W}^n]_+ + [\tilde{W}^n]_- \right) h_{\boldsymbol{\theta}_h}(\mathbf{x}_{\mathrm{adv}}) \right] + \alpha \left( [\tilde{W}^n]_+ \underline{h}_{\boldsymbol{\theta}_h}^{\mathcal{C}_\epsilon}(\mathbf{x}) + [\tilde{W}^n]_- \bar{h}_{\boldsymbol{\theta}_h}^{\mathcal{C}_\epsilon}(\mathbf{x}) \right) + \tilde{\mathbf{b}}^n \\
&= [\tilde{W}^n]_+ \left[ (1 - \alpha) h_{\boldsymbol{\theta}_h}(\mathbf{x}_{\mathrm{adv}}) + \alpha \underline{h}_{\boldsymbol{\theta}_h}^{\mathcal{C}_\epsilon}(\mathbf{x}) \right] + [\tilde{W}^n]_- \left[ (1 - \alpha) h_{\boldsymbol{\theta}_h}(\mathbf{x}_{\mathrm{adv}}) + \alpha \bar{h}_{\boldsymbol{\theta}_h}^{\mathcal{C}_\epsilon}(\mathbf{x}) \right] + \tilde{\mathbf{b}}^n.
\end{aligned}
$$

Recalling that ${}^{\mathrm{CC}}\bar{h}_{\boldsymbol{\theta}_h}^{\mathcal{C}_\epsilon}(\alpha; \mathbf{x}) := (1 - \alpha)\, h_{\boldsymbol{\theta}_h}(\mathbf{x}_{\mathrm{adv}}) + \alpha\, \bar{h}_{\boldsymbol{\theta}_h}^{\mathcal{C}_\epsilon}(\mathbf{x})$, and ${}^{\mathrm{CC}}\underline{h}_{\boldsymbol{\theta}_h}^{\mathcal{C}_\epsilon}(\alpha; \mathbf{x}) := (1 - \alpha)\, h_{\boldsymbol{\theta}_h}(\mathbf{x}_{\mathrm{adv}}) + \alpha\, \underline{h}_{\boldsymbol{\theta}_h}^{\mathcal{C}_\epsilon}(\mathbf{x})$, we get:

$$
{}^{\mathrm{CC}}\mathbf{z}_{f_{\boldsymbol{\theta}}}^{\mathcal{C}_\epsilon}(\alpha; \mathbf{x}, y) = [\tilde{W}^n]_+ {}^{\mathrm{CC}}\underline{h}_{\boldsymbol{\theta}_h}^{\mathcal{C}_\epsilon}(\alpha; \mathbf{x}) + [\tilde{W}^n]_- {}^{\mathrm{CC}}\bar{h}_{\boldsymbol{\theta}_h}^{\mathcal{C}_\epsilon}(\alpha; \mathbf{x}) + \tilde{\mathbf{b}}^n,
$$

which concludes the proof. $\qquad\square$

## C  PSEUDO-CODE

We here provide pseudo-code for CC-Dist: see algorithm 1.

---

**Algorithm 1** CC-Dist training loss $\mathcal{L}^{\text{CC-Dist}}(\alpha, {}^5\!/w; \mathbf{x}, y)$

---

1: **Input:** Student $f_{\boldsymbol{\theta}} = g_{\boldsymbol{\theta}_g} \circ h_{\boldsymbol{\theta}_h}$, teacher feature map $h_{\boldsymbol{\theta}_h^t}^t$, loss function $\mathcal{L}$, data point $(\mathbf{x}, y)$, perturbation set $\mathcal{C}_\epsilon(\mathbf{x})$, hyper-parameter $\alpha$

2: Compute $\mathbf{x}_{\text{adv}}$ via an adversarial attack on $\mathcal{C}_\epsilon(\mathbf{x})$

3: Compute $\underline{h}_{\boldsymbol{\theta}_h}^{\mathcal{C}_\epsilon}(\mathbf{x})$, $\bar{h}_{\boldsymbol{\theta}_h}^{\mathcal{C}_\epsilon}(\mathbf{x})$ and $\underline{\mathbf{z}}_{f_{\boldsymbol{\theta}}}^{\mathcal{C}_\epsilon}(\mathbf{x}, y)$ via IBP

4: ${}^{\text{CC}}\bar{h}_{\boldsymbol{\theta}_h}^{\mathcal{C}_\epsilon}(\alpha; \mathbf{x}) \leftarrow (1 - \alpha)\, h_{\boldsymbol{\theta}_h}(\mathbf{x}_{\text{adv}}) + \alpha\, \bar{h}_{\boldsymbol{\theta}_h}^{\mathcal{C}_\epsilon}(\mathbf{x})$

5: ${}^{\text{CC}}\underline{h}_{\boldsymbol{\theta}_h}^{\mathcal{C}_\epsilon}(\alpha; \mathbf{x}) \leftarrow (1 - \alpha)\, h_{\boldsymbol{\theta}_h}(\mathbf{x}_{\text{adv}}) + \alpha\, \underline{h}_{\boldsymbol{\theta}_h}^{\mathcal{C}_\epsilon}(\mathbf{x})$

6: $\mathbf{z}_{f_{\boldsymbol{\theta}}}(\mathbf{x}_{\text{adv}}, y) \leftarrow \left( g_{\boldsymbol{\theta}_g}\left( h_{\boldsymbol{\theta}_h}(\mathbf{x}_{\text{adv}}) \right)_y \mathbf{1}^k - g_{\boldsymbol{\theta}_g}\left( h_{\boldsymbol{\theta}_h}(\mathbf{x}_{\text{adv}}) \right) \right)$

7: ${}^{\text{CC}}\mathcal{L}_{f_{\boldsymbol{\theta}}}^{\mathcal{C}_\epsilon}(\alpha; \mathbf{x}, y) \leftarrow \mathcal{L}\left( -\left[ (1 - \alpha)\, \mathbf{z}_{f_{\boldsymbol{\theta}}}(\mathbf{x}_{\text{adv}}, y) + \alpha\, \underline{\mathbf{z}}_{f_{\boldsymbol{\theta}}}^{\mathcal{C}_\epsilon}(\mathbf{x}, y) \right], y \right)$

8: ${}^{\text{CC}}\mathcal{R}_{f_{\boldsymbol{\theta}}}^{\mathcal{C}_\epsilon}(\alpha; \mathbf{x}, y) \leftarrow \sum_i \max\left\{ \left( {}^{\text{CC}}\underline{h}_{\boldsymbol{\theta}_h}^{\mathcal{C}_\epsilon}(\alpha; \mathbf{x})_i - h_{\boldsymbol{\theta}_h^t}^t(\mathbf{x})_i \right)^2, \left( {}^{\text{CC}}\bar{h}_{\boldsymbol{\theta}_h}^{\mathcal{C}_\epsilon}(\alpha; \mathbf{x})_i - h_{\boldsymbol{\theta}_h^t}^t(\mathbf{x})_i \right)^2 \right\}$

9: **return** ${}^{\text{CC}}\mathcal{L}_{f_{\boldsymbol{\theta}}}^{\mathcal{C}_\epsilon}(\alpha; \mathbf{x}, y) + {}^5\!/w\ {}^{\text{CC}}\mathcal{R}_{f_{\boldsymbol{\theta}}}^{\mathcal{C}_\epsilon}(\alpha; \mathbf{x}, y)$

---

# D EXPERIMENTAL DETAILS

This appendix provides experimental details omitted from §5.

## D.1 EXPERIMENTAL SETUP

Experiments were each carried out on an internal cluster using a single GPU and employing from 6 to 12 CPU cores. GPUs belonging to the following models were employed: Nvidia GTX 1080 Ti, Nvidia RTX 2080 Ti, Nvidia RTX 6000, Nvidia RTX 8000, Nvidia V100. CPU memory was capped at 10 GB for the training experiments, and at 100 GB for the verification experiments. The runtime of training runs ranges from roughly 5 hours for CIFAR-10, to roughly 4 days for downscaled ImageNet. For verification experiments, runtime ranges from below 8 hours for CIFAR-10 with $\epsilon = {}^8\!/255$ to roughly 12 days for downscaled ImageNet.

## D.2 TRAINING SETUP AND HYPER-PARAMETERS

We now report the details of the employed training setup and hyper-parameters.

### D.2.1 DATASETS

As stated in §5.1, the experiments were carried out on the following datasets: CIFAR-10 (Krizhevsky & Hinton, 2009), TinyImageNet (Le & Yang, 2015), and downscaled ImageNet ($64 \times 64$) (Chrabaszcz et al., 2017). For all three, we train using random horizontal flips, random crops, and normalization as done in (De Palma et al., 2024b). The employed dataset splits comply with previous work in the area (De Palma et al., 2024b; Mao et al., 2023; Müller et al., 2023). Specifically, for CIFAR-10, which consists of $32 \times 32$ RGB images divided into 10 classes, we trained on the original $50,000$ training points, and evaluated on the original $10,000$ test points. For TinyImageNet, consisting of $64 \times 64$ RGB images and 200 classes, we trained on the original $100,000$ training points, and evaluated on the original $10,000$ validation images. Finally, for downscaled ImageNet, which consists of $64 \times 64$ RGB images and 1000 classes, we employed the original training set of $1,281,167$ images, and evaluated on the original $50,000$ validation images.

### D.2.2 TRAINING SCHEDULES AND IMPLEMENTATION DETAILS

For CC-IBP, CC-Dist models and their teachers, trained networks are initialized using the specialized scheme by Shi et al. (2021), which reduces the magnitude of the IBP bounds. Teachers for CC-Dist are trained using pure adversarial training, relying on a 10-step PGD attack (Madry et al., 2018) with step size $\eta = 0.25\epsilon$. Complying with De Palma et al. (2024b), $\mathbf{x}_{\text{adv}}$ for CC-Dist and CC-IBP models is computed using a single-step PGD attack with uniform random initialization and step size

$\eta = 10.0\epsilon$ on all setups except for CIFAR-10 with $\epsilon = {}^2/_{255}$, which instead relies on a 8-step PGD attack with step size $\eta = 0.25\epsilon'$ run on a larger effective input perturbation ($\epsilon' = 2.1\epsilon$). All methods use a batch size of 128 throughout the experiments, and gradient clipping with norm equal to 10. We do not perform any early stopping.

As relatively common in the adversarial training literature (Andriushchenko & Flammarion, 2020; de Jorge et al., 2022), teachers for CC-Dist are trained using the SGD optimizer (with momentum set to 0.9), shorter schedules and a cyclic learning rate, which linearly increases the learning rate from 0 to 0.2 during the first half of the training, and then decreases it back to 0. As a longer number of teacher training epochs was found to be beneficial on TinyImageNet, we report it among the hyper-parameters in table 3. As instead common in the certified training literature, CC-Dist and CC-IBP models are trained using the Adam optimizer (Kingma & Ba, 2015) with longer training schedules and a learning rate of $5 \times 10^{-4}$, which is decayed twice by a factor 0.2. Their training starts with one epoch of "warm-up", where the certified training loss is replaced by the standard cross entropy on the clean inputs, and proceeds with a "ramp-up" phase, during which the training perturbation radius $\epsilon$ is gradually increased from 0 to its target value, and the IBP regularization from Shi et al. (2021) is added to the training loss (with coefficient $\lambda = 0.5$ on CIFAR-10, and $\lambda = 0.2$ for TinyImageNet and downscaled ImageNet). Benchmark-specific details for CC-Dist models and CC-IBP, which are in accordance with De Palma et al. (2024b), follow:

- CIFAR-10 with $\epsilon = {}^2/_{255}$: the CC-IBP and CC-Dist training schedule is 160-epochs long, with 80 epochs of ramp-up, and with the learning rate decayed at epochs 120 and 140.

- CIFAR-10 with $\epsilon = {}^8/_{255}$: 260 epochs, with 80 epochs of ramp-up, and with the learning rate decayed at epochs 180 and 220.

- TinyImageNet: 160 epochs, with 80 ramp-up epochs, and decaying the learning rate at epochs 120 and 140.

- Downscaled ImageNet: 80 epochs, with 20 epochs of ramp-up, and decaying the learning rate at epochs 60 and 70.

In order to separate the effect of distillation from any other potential implementation detail, the CC-IBP evaluations from table 1 were obtained by employing the CC-Dist implementation with $\beta = 0$.

### D.2.3 NETWORK ARCHITECTURE

The employed `CNN-7` architecture is left unvaried with respect to previous work (De Palma et al., 2024b; Müller et al., 2023; Mao et al., 2023; 2025) except on downscaled ImageNet, where we applied a small modification to the linear layers. Specifically, in order to make sure that the feature space employed for distillation is larger than the network output space, we set the output size of the penultimate layer (and the input size of the last) equal to 1024, instead of 512 as for the other datasets and in De Palma et al. (2024b). As originally suggested by Shi et al. (2021) to improve performance, and complying with previous work (De Palma et al., 2024b; Müller et al., 2023; Mao et al., 2023; 2025), we employ batch normalization (BatchNorm) after every network layer except the last. In our implementation, adversarial attacks are carried out with the network in evaluation mode, hence using the current BatchNorm running statistics. Except during the warm-up phase, where we also perform an evaluation on the unperturbed data points, the network is exclusively evaluated on the computed adversarial inputs $\mathbf{x}_{\text{adv}}$, which hence dominate the computed running statistics for most of the training and for the final network. Finally, except for the clean loss during warm-up, which is computed using the unperturbed current batch statistics, all the training loss computations (including those requiring IBP bounds) are carried out using the batch statistics from the computed adversarial inputs (hence in training mode).

### D.2.4 HYPER-PARAMETERS

Table 3 reports a list of the main method and regularization hyper-parameter values for our evaluations from table 1. As done in previous work (Gowal et al., 2019; Zhang et al., 2020; Shi et al., 2021; Müller et al., 2023; De Palma et al., 2024b; Mao et al., 2025), and hence ensuring a fair comparison in table 1, tuning was carried out directly on the evaluation sets. The details of the CC-Dist and CC-IBP training schedules, taken from De Palma et al. (2024b), are instead reported in appendix D.2.2.

Table 3: Hyper-parameter settings for the CC-Dist models from table 1, their respective teachers, and for our CC-IBP evaluations reported in the same table.

| Dataset | $\epsilon$ | Method | $\alpha$ | $\ell_1$ | Teacher n. epochs | Teacher $\ell_1$ |
|---|---|---|---|---|---|---|
| CIFAR-10 | $\frac{2}{255}$ | CC-DIST | $10^{-2}$ | $3 \times 10^{-6}$ | 30 | $10^{-5}$ |
| | | CC-IBP | $10^{-2}$ | $3 \times 10^{-6}$ | / | / |
| | $\frac{8}{255}$ | CC-DIST | $0.5$ | $0$ | 30 | $5 \times 10^{-6}$ |
| | | CC-IBP | $0.5$ | $0$ | / | / |
| TinyImageNet | $\frac{1}{255}$ | CC-DIST | $5 \times 10^{-3}$ | $5 \times 10^{-5}$ | 100 | $5 \times 10^{-5}$ |
| | | CC-IBP | $3 \times 10^{-3}$ | $5 \times 10^{-5}$ | / | / |
| ImageNet64 | $\frac{1}{255}$ | CC-DIST | $5 \times 10^{-3}$ | $10^{-5}$ | 30 | $5 \times 10^{-6}$ |
| | | CC-IBP | $5 \times 10^{-3}$ | $10^{-5}$ | / | / |

As explained in §3.3, we advocate for a constant distillation coefficient: we noticed that $\beta = 5/w$ yielded good performance across our earlier CIFAR-10 experiments, and then decided to keep it to the same value across all evaluations. On CIFAR-10, we did not tune the $\alpha$ coefficient for either method: it was set to be the CC-IBP $\alpha$ coefficient employed in De Palma et al. (2024b). Complying with common practice in the certified training literature (De Palma et al., 2024b; Müller et al., 2023; Mao et al., 2025), $\ell_1$ regularization is applied on top of the employed training losses: the corresponding values for CC-Dist and CC-IBP models were not tuned but taken from the CC-IBP values reported in the expressive losses work (De Palma et al., 2024b). On TinyImageNet and ImageNet64, we found that both the standard performance and the certified accuracy of CC-IBP could be significantly improved compared to the original results from De Palma et al. (2024b), which respectively employ $\alpha = 10^{-2}$ and $\alpha = 5 \times 10^{-2}$ for CC-IBP on TinyImageNet and ImageNet64, by decreasing $\alpha$. In particular, we selected the smallest $\alpha$ value resulting in strictly better robustness-accuracy trade-offs (i.e., before certified accuracy started decreasing with lower $\alpha$ values). We trained a series of potential teachers for CC-Dist with varying $\ell_1$ coefficient and number of training epochs, and selected the teacher depending on the performance of the resulting CC-Dist model. On all datasets except TinyImageNet, where longer schedules were beneficial to CC-Dist performance, we found a teacher trained with the relatively short 30-epoch schedule to result in strong CC-Dist performance. Finally, generally speaking, we advocate for the re-use of the selected CC-IBP $\alpha$ coefficient for CC-Dist. On TinyImageNet, the only setting where the employed $\alpha$ values for the reported models differ across CC-Dist and CC-IBP, we found a larger CC-Dist $\alpha$ to result in simultaneously better standard performance and certified robustness compared to all results from the literature. Nevertheless, re-using $\alpha = 3 \times 10^{-3}$ for CC-Dist produced a model with $44.08\%$ standard accuracy and $27.40\%$ certified accuracy (compare with table 1), which, albeit not improving on both metrics at once compared to the MTL-IBP results from (Mao et al., 2025), significantly improves upon the overall performance trade-offs seen in the literature.

Unless otherwise stated, all hyper-parameters throughout the experimental evaluations comply with those reported in table 3.

### D.3 VERIFICATION SETUP

The employed verification setup is analogous to the one from De Palma et al. (2024b). As stated in §5, we use the open source OVAL verification framework (Bunel et al., 2018; 2020a; De Palma et al., 2021), which performs branch-and-bound, and a configuration based on alpha-beta-CROWN bounds (Wang et al., 2021). Before running branch-and-bound, we try verifying the property via IBP and CROWN bounds from `auto_LiRPA` (Xu et al., 2020), or to falsify it through a PGD attack (Madry et al., 2018). Differently from De Palma et al. (2024b), we use a timeout of 600 seconds (as opposed to 1800 seconds). We perform verification by running the framework to compute $\min_i \mathbf{z}_{f_\theta}^{\mathcal{C}_\epsilon}(\mathbf{x}, y)_i$, where the $\min$ operator is converted into an equivalent auxiliary ReLU network to append to $\mathbf{z}_{f_\theta}(\mathbf{x}, y)$ (Bunel et al., 2020b; De Palma et al., 2024b). For TinyImageNet and downscaled ImageNet, differently from De Palma et al. (2024b), in order to reduce the size of the auxiliary network, we exclude from the $\min$ operator all the logit differences that were already proved to be positive by the IBP or CROWN bounds computed before running branch-and-bound. Finally, the configuration employed to verify the TinyImageNet and downscaled ImageNet teacher models

(results reported in table 2) computes looser pre-activation bounds (using CROWN (Zhang et al., 2018) as opposed to 5 iterations of alpha-CROWN (Xu et al., 2021)) at the root of branch-and-bound: this was found to be a more effective verifier for networks trained via pure adversarial training.

### D.4 SOFTWARE ACKNOWLEDGMENTS AND LICENSES

As described above, our code relies on the OVAL verification framework to verify the models, which was released under an MIT license. The training codebase is analogous to the one from the expressive losses work (De Palma et al., 2024b), also released under an MIT license: this was in turn based on the codebase from Shi et al. (2021), released under a 3-Clause BSD license. Both above repositories, and hence our codebase, rely on the `auto_LiRPA` (Xu et al., 2020) framework for incomplete verification, which has a 3-Clause BSD license. Concerning datasets: downscaled ImageNet was obtained from the ImageNet website (`https://www.image-net.org/download.php`), and TinyImageNet from the website of the CS231n Stanford class (`http://cs231n.stanford.edu/TinyImageNet-200.zip`). MNIST and CIFAR-10 were instead downloaded using `torchvision.datasets` (Paszke et al., 2019).

## E SUPPLEMENTARY EXPERIMENTS

This appendix reports experimental results omitted from the main paper.

### E.1 EFFECT OF DISTILLATION

Aiming to shed further light behind the effect of knowledge distillation in this context, we here present a more detailed experimental comparison between the CC-IBP and CC-Dist models from table 1, presenting AutoAttack (Croce & Hein, 2020) accuracy to measure empirical robustness, and the IBP loss to measure the ease of verifiability. As we can conclude from table 4, the use of knowledge distillation improves trade-offs between standard accuracy and certified robustness by:

1. transferring some of the superior natural accuracy and empirical robustness of the teacher onto the student through the distillation term;

2. preserving a large degree of verifiability through the CC-IBP term within the CC-Dist loss.

Table 4: Detailed analysis of the effect on distillation onto CC-IBP.

| Dataset | $\epsilon$ | Method | Standard acc. [%] | AA acc. [%] | Certified acc. [%] | IBP loss |
|---|---|---|---|---|---|---|
| CIFAR-10 | $\frac{2}{255}$ | CC-DIST | 81.55 | 70.71 | 64.60 | 29.05 |
| | | CC-IBP | 79.51 | 68.56 | 63.50 | 27.46 |
| | $\frac{8}{255}$ | CC-DIST | 55.13 | 36.80 | 35.52 | 1.865 |
| | | CC-IBP | 54.46 | 36.59 | 35.42 | 1.862 |
| TinyImageNet | $\frac{1}{255}$ | CC-DIST | 43.78 | 32.30 | 27.88 | 59.63 |
| | | CC-IBP | 41.28 | 30.12 | 26.53 | 39.39 |
| ImageNet64 | $\frac{1}{255}$ | CC-DIST | 28.17 | 17.87 | 13.96 | 50.46 |
| | | CC-IBP | 25.94 | 16.51 | 13.69 | 30.66 |

### E.2 OTHER SPECIAL CASES OF THE PROPOSED DISTILLATION LOSS

As seen in proposition 3.2, when choosing $\alpha \in [0, 1]$, the proposed distillation loss ${}^{\mathrm{CC}}\mathcal{R}_{f_{\boldsymbol{\theta}}}^{\mathcal{C}_{\epsilon}}(\alpha; \mathbf{x}, y)$ can continuously interpolate between lower and upper bounds to the worst-case feature-space distillation loss in equation (8). While §3.3 advocates for the use of the same $\alpha$ coefficient employed by CC-IBP in order to closely mirror its loss (see lemma 3.3), we expect other choices of the $\alpha$ coefficient to work well on selected benchmarks. We here evaluate the behavior of two training schemes that correspond to other special cases of the proposed parametrized distillation loss. Specifically, we consider the use of ${}^{\mathrm{CC}}\mathcal{R}_{f_{\boldsymbol{\theta}}}^{\mathcal{C}_{\epsilon}}(0; \mathbf{x}, y) = \mathcal{R}_{f_{\boldsymbol{\theta}}}^{\mathcal{C}_{\epsilon}}(\mathbf{x}, y)$ and ${}^{\mathrm{CC}}\mathcal{R}_{f_{\boldsymbol{\theta}}}^{\mathcal{C}_{\epsilon}}(1; \mathbf{x}, y) = \bar{\mathcal{R}}_{f_{\boldsymbol{\theta}}}^{\mathcal{C}_{\epsilon}}(\mathbf{x}, y)$ on top of CC-IBP,

calling the resulting training schemes CC-Dist$_0$ and CC-Dist$_1$, respectively:

$$\mathcal{L}^{\text{CC-Dist}_0}(\alpha, \beta; \mathbf{x}, y) := {}^{\text{CC}}\mathcal{L}_{f_\theta}^{\mathcal{C}_\epsilon}(\alpha; \mathbf{x}, y) + \beta \ {}^{\text{CC}}\mathcal{R}_{f_\theta}^{\mathcal{C}_\epsilon}(0; \mathbf{x}, y),$$

$$\mathcal{L}^{\text{CC-Dist}_1}(\alpha, \beta; \mathbf{x}, y) := {}^{\text{CC}}\mathcal{L}_{f_\theta}^{\mathcal{C}_\epsilon}(\alpha; \mathbf{x}, y) + \beta \ {}^{\text{CC}}\mathcal{R}_{f_\theta}^{\mathcal{C}_\epsilon}(1; \mathbf{x}, y).$$

Keeping $\ell_1$ regularization and the $\alpha$ coefficient of the CC-IBP loss fixed, we first tested different distillation coefficients $\beta$ while using the teachers employed for the CC-Dist models (see tables 2 and 3), finding that $\beta = 5/w$ yields the largest natural accuracy improvement for both methods on $\epsilon = 8/255$, and a strong trade-off between certified robustness and standard performance for CC-Dist$_0$ on $\epsilon = 2/255$. We found $\beta = (5 \times 10^{-4})/w$ to work better for CC-Dist$_1$ on $\epsilon = 2/255$. Then, using the above distillation coefficients, and similarly to what done for CC-Dist (see appendix D.2.4), we tested the two methods using teachers with varying $\ell_1$ regularization and trained for 30 epochs. Aiming to compare them with CC-Dist, Table 5 reports results for the best-performing CC-Dist$_0$ and CC-Dist$_1$ models from the above procedure that display similar performance profiles with CC-Dist.

Table 5 shows that both CC-Dist$_0$ and CC-Dist$_1$ work well across the considered CIFAR-10 settings. As we would expect considering the low employed $\alpha$ coefficient (see table 3), CC-Dist$_0$ strictly outperforms CC-Dist$_1$ on $\epsilon = 2/255$, where both methods nevertheless improve on both the standard performance and the certified accuracy of CC-IBP. While CC-Dist$_0$ displays performance comparable to CC-Dist in this setting, CC-Dist$_1$ performs markedly worse. On $\epsilon = 8/255$, where the employed $\alpha$ coefficient is instead relatively large (see table 3), CC-Dist$_1$ strictly outperforms CC-Dist$_0$ on both reported performance profiles, indicating the benefit of distilling onto feature IBP bounds in this context. Nevertheless, both methods can produce models improving on CC-IBP for both metrics on $\epsilon = 8/255$, with the corresponding CC-Dist$_0$ model strictly outperformed by CC-Dist.

We believe that the above results further confirm the effectiveness of using empirically-robust models to improve certified training via knowledge distillation. As the relative performance of CC-Dist$_0$ and CC-Dist$_1$ varies depending on the experimental setting, we advocate for the use of CC-Dist as described in §3.3 due to its adaptive nature. Nevertheless, given that most considered benchmarks require relatively low $\alpha$ coefficients (see table 3), we expect CC-Dist$_0$ to be a valid alternative to CC-Dist on TinyImageNet and downscaled ImageNet.

### E.3    DISTILLATION ONTO OTHER CERTIFIED TRAINING SCHEMES

In order to showcase the wider applicability of knowledge distillation from empirically-robust teachers, we here present results on applying such distillation process onto other certified training schemes, focusing on SABR (Müller et al., 2023), MTL-IBP (De Palma et al., 2024b), and IBP-R (De Palma et al., 2022). While the first two algorithms are expressive losses (De Palma et al., 2024b) like CC-IBP, which is the focus of this paper, IBP-R does not fit into the relative framework. As described in §4, SABR computes IBP bounds over a subset of $\mathcal{C}_\epsilon$, termed a "small box". We propose to employ a distillation loss of the form of ${}^{\text{CC}}\mathcal{R}_{f_\theta}^{\mathcal{C}_\epsilon}(1; \mathbf{x}, y)$, yet using IBP bounds computed on the small box, calling the resulting algorithm SABR-Dist. For MTL-IBP, which takes convex combinations between $\mathcal{L}_{f_\theta}^{\mathcal{C}_\epsilon}(\mathbf{x}, y)$ and $\bar{\mathcal{L}}_{f_\theta}^{\mathcal{C}_\epsilon}(\mathbf{x}, y)$, we employ the same distillation loss used for CC-Dist: ${}^{\text{CC}}\mathcal{R}_{f_\theta}^{\mathcal{C}_\epsilon}(1; \mathbf{x}, y)$.

Table 5: Comparison between the CC-Dist models from table 1 and other special cases of the proposed parametrized distillation loss.

| Dataset | $\epsilon$ | Method | $\beta$ | Teacher $\ell_1$ | Standard acc. [%] | Certified acc. [%] |
|---|---|---|---|---|---|---|
| CIFAR-10 | $\frac{2}{255}$ | CC-DIST | $5/w$ | $10^{-5}$ | 81.55 | 64.60 |
| | | CC-DIST$_0$ | $5/w$ | $10^{-5}$ | 81.74 | 64.22 |
| | | CC-DIST$_1$ | $(5 \times 10^{-4})/w$ | $5 \times 10^{-5}$ | 79.84 | 63.93 |
| | | CC-IBP | 0 | / | 79.51 | 63.50 |
| | $\frac{8}{255}$ | CC-DIST | $5/w$ | $5 \times 10^{-6}$ | 55.13 | 35.52 |
| | | CC-DIST$_0$ | $5/w$ | $2 \times 10^{-5}$ | 55.63 | 35.08 |
| | | CC-DIST$_1$ | $5/w$ | $2 \times 10^{-6}$ | 55.91 | 35.35 |
| | | CC-DIST$_0$ | $2/w$ | $5 \times 10^{-6}$ | 54.65 | 35.46 |
| | | CC-DIST$_1$ | $2/w$ | $5 \times 10^{-6}$ | 54.87 | 35.55 |
| | | CC-IBP | 0 | / | 54.46 | 35.42 |

Table 6: Effect of distillation from empirically-robust teachers onto SABR (Müller et al., 2023), MTL-IBP (De Palma et al., 2024b), and IBP-R (De Palma et al., 2022).

| Dataset | $\epsilon$ | Method | Std. acc. [%] | PGD-40 acc. [%] | Cert. acc. [%] |
|---------|-----------|--------|---------------|-----------------|----------------|
| CIFAR-10 | $\frac{2}{255}$ | SABR-Dist | 81.18 | 71.13 | 64.54 |
| | | SABR | 79.55 | 69.49 | 63.94 |
| | | MTL-IBP-Dist | 81.58 | 71.88 | 64.09 |
| | | MTL-IBP | 79.71 | 69.67 | 63.16 |
| | | IBP-R-Dist | 81.13 | 70.28 | 61.76 |
| | | IBP-R | 79.68 | 69.98 | 61.43 |

Similarly to CC-Dist, and in spite of the lack of the coupling provided by lemma 3.2, we re-use the same $\alpha$ coefficient as for MTL-IBP. We note, nevertheless, that, for any given $\alpha$, the MTL-IBP loss upper bounds the CC-IBP loss (De Palma et al., 2024b, proposition 4.2). For IBP-R, we instead use $^{\mathrm{CC}}\mathcal{R}_{f_{\theta}}^{\mathcal{C}_{\epsilon}}(0; \mathbf{x}, y)$ as distillation loss (see appendix E.2). Various teachers and distillation coefficients were tested for IBP-R, whereas we only varied $\beta$ and kept the same teacher as table 2 for SABR and MTL-IBP. $\beta = {}^{5}/w$ was chosen for all methods. For IBP-R a teacher with $\ell_1$ coefficient of $2 \times 10^{-5}$ was selected. Table 6 shows that the distillation process successfully improves both the standard accuracy and the certified robustness of all the three considered methods, demonstrating the wider potential of distillation from adversarially-trained teachers towards certified training.

### E.4 LOGIT-BASED DISTILLATION

We now compare the results of our proposed distillation loss from §3.2 with a loss that instead seeks to directly distill onto the CC-IBP convex combinations $^{\mathrm{CC}}\mathbf{z}_{f_{\theta}}^{\mathcal{C}_{\epsilon}}(\alpha; \mathbf{x}, y)$. Specifically, it employs a KL term between $^{\mathrm{CC}}\mathbf{z}_{f_{\theta}}^{\mathcal{C}_{\epsilon}}(\alpha; \mathbf{x}, y)$ and the teacher logit differences $\mathbf{z}_{t_{\theta t}}(\mathbf{x}, y)$, resulting in the following training loss:

$$^{\mathrm{CC}}\mathcal{L}_{f_{\theta}}^{\mathcal{C}_{\epsilon}}(\alpha; \mathbf{x}, y) + T^2 \beta \, \mathrm{KL}_T \left( -{}^{\mathrm{CC}}\mathbf{z}_{f_{\theta}}^{\mathcal{C}_{\epsilon}}(\alpha; \mathbf{x}, y), -\mathbf{z}_{t_{\theta t}}(\mathbf{x}, y) \right). \tag{15}$$

Keeping all other hyper-parameters fixed (including the teacher model) to the values reported in table 3, we tested different distillation coefficients $\beta$, using $T = 20$. We found that, compared to CC-IBP, the best performance profile using equation (15) were obtained using $\beta = {(10^4)}/w$ for both considered CIFAR-10 setups, whose results are reported in table 7. CC-Dist outperforms the logit-space distillation loss in both settings. We ascribe this to the more informative content of the model features, and to the inherent difference between the training goals of the teacher and the student, which are respectively trained for empirical and certified adversarial robustness. Allowing the student to learn a markedly different classification head from the teacher may be beneficial in this context.

### E.5 PREACTRESNET18 TEACHERS

We now present a preliminary evaluation of the effect of employing a PreActResNet18 (PRN18) architecture (He et al., 2016) as teacher for CC-Dist. In order to ensure that the feature space of the teacher is set to the output of a ReLU, in compliance with the student model, we place the PreActResNet18 average pooling layer before the last ReLU and batch normalization (the standard PreActResNet18 architecture places it instead before the last linear layer). Focusing on TinyImageNet, and keeping $\alpha = 3 \times 10^{-3}$ and the $\ell_1$ coefficient to $5 \times 10^{-5}$ as for the CC-IBP model on this setup (see table 3), we tested various PRN18 teachers trained for 30 epochs with varying $\ell_1$ regularization as CC-Dist teachers. Table 8 compares the performance of the ensuing CC-Dist model having the

Table 7: Comparison between the logit-space distillation from equation (15) and the CC-Dist models from table 1.

| Dataset | $\epsilon$ | Method | Standard acc. [%] | Certified acc. [%] |
|---------|-----------|--------|-------------------|--------------------|
| CIFAR-10 | $\frac{2}{255}$ | CC-DIST | 81.55 | 64.60 |
| | | EQ. (15) | 80.61 | 63.65 |
| | $\frac{8}{255}$ | CC-DIST | 55.13 | 35.52 |
| | | EQ. (15) | 54.76 | 35.20 |

Table 8: Evaluation of the effect of PreActResNet18 (`PRN18`) teachers (tch.) on the TinyImageNet performance of CC-Dist, compared to teachers with the same architecture as the student.

| Dataset | $\epsilon$ | Tch. arch. | Std. acc. [%] | Cert. acc. [%] | Tch. std. acc. [%] | Tch. PGD-40 acc. [%] |
|---|---|---|---|---|---|---|
| TinyImageNet | $\frac{1}{255}$ | `CNN-7` | 44.08 | 27.40 | 47.18 | 36.07 |
| | | `PRN18` | 43.03 | 26.09 | 50.90 | 39.98 |

largest natural accuracy with the best model obtained through same-architecture teachers for these hyper-parameters, and the respective teachers. CC-Dist draws no benefit from the `PRN18` teacher (which was trained with $\ell_1$ coefficient equal to $5 \times 10^{-5}$), in spite of the fact that it displays stronger performance than the `CNN-7` teacher. While we are hopeful that better CC-Dist results could be obtained by training `PRN18` teachers for longer (the considered teachers are trained for 30 epochs, as opposed to the 100 epochs employed for the `CNN-7` teacher), we leave this for future work.

## E.6 EXPERIMENTAL VARIABILITY

Owing to the large cost of repeatedly training and verifying certifiably-robust models (the worst-case per-image verification runtime is 600 seconds: see appendices D.1 and D.3), and as common in the area (De Palma et al., 2024b; Mao et al., 2023; Müller et al., 2023; Mao et al., 2024a; 2025), all the experiments were run using a single seed. In order to provide an indication of experimental variability, table 9 presents aggregated CIFAR-10 results over 4 repetitions for CC-Dist and CC-IBP. These include the CC-Dist and CC-IBP results reported in table 1 and 3 further repetitions of the associated experiment, consisting of training and the ensuing verification using branch-and-bound. We found experimental variability to be relatively low on $\epsilon = 2/255$, and more noticeable on $\epsilon = 8/255$. On the latter setting, distillation markedly improves the average standard accuracy while leaving certified robustness roughly unvaried. For $\epsilon = 2/255$, CC-Dist instead produces a significant improvement on both average metrics at once. In both cases, the cumulative (signed) improvement across the two averaged metrics is similar to the one reported in table 1.

## E.7 RUNTIME MEASUREMENTS

In order to assess the training overhead associated with our distillation scheme, we here present runtime measurements and estimates for CC-Dist and CC-IBP. Specifically, we provide the training runtime of both methods, separately including also the teacher training runtime for CC-Dist, and estimates of the verification runtimes for the trained models from table 1. These experiments were carried out using an Nvidia RTX 8000 GPU, and 6 cores of an AMD EPIC 7302 CPU. Table 10 shows that, under the training schedules of appendix D.2.2 and when training teachers for 30 epochs as per table D.2.4, CC-Dist is associated with minimal training overhead on the considered CIFAR-10 settings. This overhead increases to respectively almost 70% and 40% for TinyImageNet and down-scaled ImageNet, where stronger teachers are required in order to maximize performance. Finally, the increased certified accuracy of the CC-Dist models (see table 1) comes with an increased verification runtime, as expected from the increases IBP loss resulting from the distillation process (see table 4).

Table 9: Experimental variability of CC-Dist and CC-IBP on CIFAR-10: maximal and minimal values, the mean and its standard error (SEM) across 4 repetitions are reported.

| Dataset | $\epsilon$ | Method | Standard acc. [%] | | | | Certified acc. [%] | | | |
|---|---|---|---|---|---|---|---|---|---|---|
| | | | Mean | SEM | Max | Min | Mean | SEM | Max | Min |
| CIFAR-10 | $\frac{2}{255}$ | CC-DIST | 81.58 | 0.08 | 81.72 | 81.38 | 64.46 | 0.14 | 64.61 | 64.03 |
| | | CC-IBP | 79.68 | 0.08 | 79.90 | 79.51 | 63.41 | 0.10 | 63.65 | 63.21 |
| | $\frac{8}{255}$ | CC-DIST | 55.22 | 0.07 | 55.40 | 55.09 | 34.88 | 0.24 | 35.52 | 34.40 |
| | | CC-IBP | 54.12 | 0.19 | 54.62 | 53.73 | 34.89 | 0.23 | 35.42 | 34.34 |

Table 10: Training runtime measurements for CC-Dist, their respective teachers (trained for 30 epochs), and CC-IBP under the training schedules of appendix D.2.2.

| Dataset | $\epsilon$ | Method | Training runtime [s] | Teacher training runtime [s] | Estimated[†] verification runtime [s] |
|---------|-----------|--------|---------------------|------------------------------|--------------------------------------|
| CIFAR-10 | $\frac{2}{255}$ | CC-DIST | $1.605 \times 10^4$ | $2.225 \times 10^3$ | $1.097 \times 10^5$ |
|  |  | CC-IBP | $1.568 \times 10^4$ | / | $7.089 \times 10^4$ |
|  | $\frac{8}{255}$ | CC-DIST | $1.653 \times 10^4$ | $2.223 \times 10^3$ | $1.090 \times 10^4$ |
|  |  | CC-IBP | $1.600 \times 10^4$ | / | $6.238 \times 10^3$ |
| TinyImageNet | $\frac{1}{255}$ | CC-DIST | $5.432 \times 10^4$ | $3.470 \times 10^4$ | $2.778 \times 10^5$ |
|  |  | CC-IBP | $5.246 \times 10^4$ | / | $1.608 \times 10^5$ |
| ImageNet64 | $\frac{1}{255}$ | CC-DIST | $3.228 \times 10^5$ | $1.089 \times 10^5$ | $1.210 \times 10^6$ |
|  |  | CC-IBP | $3.095 \times 10^5$ | / | $7.794 \times 10^5$ |

[†]Extrapolated from measurements over the first 500 test images.

## E.8 DIFFERENT ARCHITECTURE AND ACTIVATION FUNCTION

The main results focus on the `CNN-7` architecture owing to its state-of-the-art performance and prevalence in the relevant literature (De Palma et al., 2024b; Mao et al., 2023; Müller et al., 2023; Shi et al., 2021). We here investigate whether the distillation process is beneficial beyond this context by studying the relative performance of CC-Dist on a different activation function and on a different architecture for both the teacher and the model, focussing on CIFAR-10 with $\epsilon = 2/255$. For the activation experiment, we modify `CNN-7` to employ the hyperbolic tangent (tanh), testing different teachers trained for 100 epochs (choosing a teacher $\ell_1$ coefficient of $2 \times 10^{-5}$), and varying $\beta$ values (settling for $\beta = 5/w$ as for the experiments in table 1). For the architecture experiment, we use PreActResNet18 (`PRN18`) for both the teacher and the student, testing different teachers trained for 30 epochs (settling on a teacher with $\ell_1$ coefficient of $5 \times 10^{-6}$), and using $\beta = 5/w$. In both cases, owing to the lack of support for either model from the OVAL branch-and-bound framework (Bunel et al., 2018; 2020a; De Palma et al., 2021) employed throughout the paper, we use CROWN (Zhang et al., 2018) from `auto_LiRPA` (Xu et al., 2020) as post-training verification algorithm. Table 11 shows that CC-Dist successfully improves robustness-accuracy trade-offs for both the considered settings, demonstrating the wider applicability of the proposed distillation technique.

## E.9 DISTILLATION ON EARLIER LATENTS

Throughout the paper, we define the feature space as the activations before last affine network layer, in accordance with the conditions of lemma 3.2. We here investigate the effect of computing on the distillation loss ${}^{\text{CC}}\mathcal{R}^{\mathcal{C}_\epsilon}_{f_\theta}(\alpha; \mathbf{x}, y)$ on an earlier feature space, corresponding to the activations before the penultimate affine layer of `CNN-7`. In particular, we focus on CIFAR-10 with $\epsilon = 2/255$, keeping the teacher model fixed to the one used for tables 1 and 2 and varying the $\beta$ coefficient to account for the change in the feature space: we found $\beta = 2/w$ to yield the maximize performance in this context. Table 12 shows that distillation for the last affine layer yields strictly better robustness-accuracy trade-offs than those presented in table 5.1. As distilling on earlier layers implies that a larger portion of the network is exclusively trained using the CC-IBP loss, we ascribe this to insufficient teacher-student coupling.

## E.10 CLEAN TEACHERS

In spite of our focus on learning from adversarially-trained teachers, the proposed distillation loss ${}^{\text{CC}}\mathcal{R}^{\mathcal{C}_\epsilon}_{f_\theta}(\alpha; \mathbf{x}, y)$ only makes use of the clean features of the teacher model. In order to investigate

Table 11: Effect of distillation beyond `CNN-7` students on CIFAR-10 with $\epsilon = 2/255$.

(a) Modified `CNN-7` with tanh activation functions.

| Method | Std. acc. [%] | CROWN acc. [%] |
|--------|---------------|----------------|
| CC-Dist | 72.38 | 49.42 |
| CC-IBP | 71.29 | 47.39 |

(b) PreActResNet18 architecture.

| Method | Std. acc. [%] | CROWN acc. [%] |
|--------|---------------|----------------|
| CC-Dist | 74.01 | 53.63 |
| CC-IBP | 73.40 | 52.88 |

Table 12: Effect of computing the distillation loss $^{\text{CC}}\mathcal{R}_{f_{\boldsymbol{\theta}}}^{\mathcal{C}_\epsilon}(\alpha; \mathbf{x}, y)$ on an earlier feature space.

| Dataset | $\epsilon$ | Feature space | Std. acc. [%] | Cert. acc. [%] |
|---------|-----------|---------------|---------------|----------------|
| CIFAR-10 | $\frac{2}{255}$ | Last | 81.55 | 64.60 |
| | | Penultimate | 80.84 | 63.11 |

whether the empirical robustness of the teacher is indeed necessary for an effective distillation process, we here study the effect of employing standard-trained teachers. In particular, focusing on CIFAR-10 with $\epsilon = 2/255$, we keep $\beta = 2/w$ and test the use of SGD-trained teachers trained with varying $\ell_1$ regularization coefficients ($5 \times 10^{-6}$ being the chosen $\ell_1$ coefficient). Table 13 shows that the use of standard teachers markedly worsens the certified accuracy of the student, which is now inferior to the one associated to CC-IBP (see table 1). These results demonstrate that a robust teacher representation is a key requirement for the success of the distillation process.

# F    DETAILS ON THE EMPLOYED NOTATION

We here provide additional details on the employed notation.

**Student and teacher models**    Input-to-logits maps are denoted using the letter $f$, defined as the composition of a feature map, denoted using the letter $h$, and classification heads, denoted using the letter $g$. We employ a subscript to denote the parameters of these (sub-)networks, here written as a function of their inputs only. Network parameters, denoted $\boldsymbol{\theta}$ throughout this work, are subscripted to denote the subsets of $\boldsymbol{\theta}$ corresponding to the feature map $\boldsymbol{\theta}_h$, and to the classification head, $\boldsymbol{\theta}_g$, respectively. The (sub-)networks for the teacher model, and their parameters, are denoted by the use of the $t$ superscript (e.g, $\boldsymbol{\theta}_h^t$ or $h_{\boldsymbol{\theta}_h^t}^t$).

**Bounds to worst-case quantities**    The worst-case classification loss, equation (3), and the worst-case distillation loss, equation (8), are superscripted by the relative local perturbation set $\mathcal{C}_\epsilon$ around the input $\mathbf{x}$. Lower and upper bounds to both quantities are denoted through lower and upper bars, respectively (e.g., $\mathcal{L}_{f_{\boldsymbol{\theta}}}^{\mathcal{C}_\epsilon}(\mathbf{x}, y)$), with the superscript preserved to stress their local validity. A similar notation is employed for the local bounds to the worst-case logit differences (e.g, $\mathbf{z}_{f_{\boldsymbol{\theta}}}^{\mathcal{C}_\epsilon}(\mathbf{x}, y)$, which lower bounds equation 2. Convex combinations between lower and upper bounds, parametrized by $\alpha$, are denoted by left-superscript $CC$ standing for Convex Combination (e.g., $^{\text{CC}}\mathcal{R}_{f_{\boldsymbol{\theta}}}^{\mathcal{C}_\epsilon}(\alpha; \mathbf{x}, y)$), without any bars. For all these worst-case quantities, lower bounds corresponding to evaluations on concrete inputs from adversarial attacks, and the upper bounds are obtained through network convex relaxations (specifically, IBP).

**Bounds to the student features**    We again employ a similar notation when bounding the student features $h_{\boldsymbol{\theta}_h}(\mathbf{x})$, defined in proposition 3.1. In this context, both the lower and the upper bounds are obtained through IBP. Convex combinations between the adversarial student latents $h_{\boldsymbol{\theta}_h}(\mathbf{x}_{\text{adv}})$ and the IBP lower and upper bounds, as defined in equation 9, are denoted by the left-superscript $CC$ and lower and upper bars, respectively (e.g., $^{\text{CC}}\bar{h}_{\boldsymbol{\theta}_h}^{\mathcal{C}_\epsilon}(\alpha; \mathbf{x})$).

Table 13: Effect of performing distillation from clean teachers.

| Dataset | $\epsilon$ | Teacher training | Std. acc. [%] | Cert. acc. [%] |
|---------|-----------|------------------|---------------|----------------|
| CIFAR-10 | $\frac{2}{255}$ | PGD-10 | 81.55 | 64.60 |
| | | Standard | 81.48 | 62.52 |

