# OpenReview forum: "Learning Better Certified Models from Empirically-Robust Teachers"
_ICLR.cc/2026/Conference — Submitted to ICLR 2026_

### Official Review · Reviewer_DdPg · 2025-10-19

**Soundness:** 3
**Presentation:** 3
**Contribution:** 3
**Rating:** 8
**Confidence:** 5

**Summary:**

This paper studies improving the accuracy-robustness tradeoff in certified training via transferring knowledge from an adversarially trained teacher. While the idea is simple, this work proposes clean and computationally efficient ways to combine the soft-label of the teacher model into the training loss. Basically, they regularize the IBP bounds in the feature space to be close to the feature of the adversarial model.

**Strengths:**

The idea is clean and effective. The proposed algorithm is efficient and integrates nicely into one of the SOTA certified training algorithm. The paper is well-written. The improvement on the clean accuracy is significant, with nontrivial improvement on the certified robustness.

**Weaknesses:**

The algorithm is heavily dependent on CC-IBP, one of the SOTA algorithms. It is not clear how to combine the idea of knowledge transfer into other SOTA algorithms, even similar ones such as MTL-IBP. This might limit the extension of the idea, while at a simple glance all certified training algorithms should benefit from knowledge transfer.

Some related work regarding certified training is missing: [1] studies the optimization difficulty of precise convex relaxations; [2] proves there exists no network encoding a simple function that may be found via certified training where a single convex relaxation such as IBP can provide precise bounds; [3] further shows that there exists certain network encoding every piecewise linear functions that may be found via certified training where layerwise multi-neuron relaxations can provide precise bounds.

[1] https://arxiv.org/abs/2403.07095

[2] https://arxiv.org/abs/2311.04015

[3] https://arxiv.org/abs/2410.06816

**Questions:**

Besides the points raised in the weakness, please address the following technical questions.

1. In Line 200, the loss controls the  deviation between the feature of the student model on the worst-case input and the **clean** feature of the teacher model. Therefore, it seems the teacher model only needs to be good at the clean features. Could the authors then show why an adversarially robust teacher might be necessary in good performance (if this is the case by intuition and the default method). Experimental comparisons are required.

---

> ### Author Response · Authors · 2025-11-23
>
> We sincerely thank Reviewer DdPg for their strong positive evaluation, and for the valuable question and observations, which have helped us increase the quality of our submission.
>
> We are particularly happy that the reviewer found our idea to be clean and effective, and the relative performance improvements to be significant.
> We will now individually address the reported weaknesses and questions. Requested changes are highlighted in blue in the revision.
>
> > *is not clear how to combine the idea of knowledge transfer into other SOTA algorithms, even similar ones such as MTL-IBP.*
>
> Thank you for your observation. We absolutely agree that in principle all certified training algorithms should benefit from knowledge transfer.
> Appendix E.3, which used to demonstrate that the distillation process effectively generalizes onto SABR, has now been extended to include MTL-IBP and IBP-R experiments.
> We adopt the following distillation losses:
> - for SABR, we distill onto the SABR bounds (the IBP bounds on the small boxes);
> - for MTL-IBP, we use the same distillation loss as CC-Dist (using the same $\alpha$ as for the MTL-IBP loss);
> - for IBP-R, we distill onto the adversarial features/
> In all three cases, distillation resulted in better trade-offs between certified and standard accuracy, further highlighting the potential of knowledge distillation in the area.
>
> > *Some related work regarding certified training is missing*
>
> Thank you for pointing this out: the related work section has been amended to include the provided references.
>
> > *it seems the teacher model only needs to be good at the clean features. Could the authors then show why an adversarially robust teacher might be necessary in good performance*
>
> We thank the reviewer for the question.
> While, indeed, only the clean teacher features appear in the distillation loss, the teacher's robustness plays a fundamental role in the overall success of the distillation strategy.
> In order to shed light on the importance of an adversarially-trained teacher for CC-Dist, the newly-added appendix E.10 investigates the use of SGD-trained teachers.
> Table 13 shows that distillation from clean teachers results in markedly worse performance profiles, highlighting the importance of teaching from robust representations.
> In particular, while a clean teacher manages to increase the student's standard accuracy, it does so at the expense of certified robustness. In other words, an inherently vulnerable representation steers the student away from robustness.

---

> > ### Comment · Reviewer_DdPg · 2025-11-26
> >
> > Thanks for the detailed reply. My concerns are addressed.

---

### Official Review · Reviewer_S6Q6 · 2025-10-30

**Soundness:** 2
**Presentation:** 2
**Contribution:** 2
**Rating:** 2
**Confidence:** 3

**Summary:**

The problem this work is trying to solve is to improve the standard performance of a certifiably robust model, so as to match an adversarially trained model. The authors propose the CC-DIST that consists of two terms. The first term is an interpolation between the upper bound (loss at the adversarial example) and the lower bound (certified loss) of the adversarial loss. The second term is a distillation loss between the adversarial bound (with interpolation) of the internal representation of the student model, and the clean representation of the teacher model. The authors tested their method on some commonly used datasets.

**Strengths:**

1. Section 2 is a nice description of the background
2. The experiment section includes a number of experiments

**Weaknesses:**

I think there are three main weaknesses: presentation, motivation, and experiments.

## 1. Presentation
The presentation could be greatly improved. Even though I am pretty familiar with the subject, I feel overwhelmed by the notations in this manuscript that look very similar to one another. I cannot see why it is necessary to add CC as a subscript before everything, and those $\theta_h$, $\theta_h^t$ etc. are all very confusing.

I think the idea is pretty simple. Basically there are two ways to get bounds: the empirical bound obtained from an adversarial example, and the certified bound obtained from IBP. So why not just use $h(x_{adv})$ to denote the empirical bound, $h$ with upper and lower bars to denote the certified bound, and use $h_t$ for the student model? The current notations are very hard to read.

For section 3, I think that the idea can be better demonstrated with diagrams. For example, the definition of the interpolations, the distillation loss, etc. I also don't see why Lemma 3.3 is important.

Regarding the experiment section, there are an unnecessarily large number of compared methods, but some important parts are missing (such as how $\alpha$ is selected), which I will explain in more detail later.

## 2. Motivation
The motivation of this work is to give certified robust models the same standard accuracy as adversarially trained models. First, this is very confusing because adversarially trained models have much lower accuracy than vanilla training. Second, I cannot see how the different components in the proposed method fulfill this motivation. For example, why interpolating between the empirical and the certified bounds? Why distilling from a teacher model guarantees that the student model will have both better standard and certified adversarial performances, instead of both worse?

In other words, I find the connection between the method and the motivation very weak, and the motivation itself is questionable. I am not even sure if the central problem studied in this work is a valid one.

## 3. Experiment
Finally, the experiment section does not show strong evidence that the proposed method is doing anything interesting. Surely Table 1 shows that the proposed method, as the authors tested it, has higher standard and certified accuracy than the compared methods. But there are several issues.
1. The reported results are not significantly better
2. The experiment setting is very unclear. For example, I cannot find in the paper how $\alpha$ is selected in the experiments, which seems to me to be very important
3. The results for previous methods are reported in prior work, and it is unclear whether the improvement of the proposed method comes from a different evaluation setting. The authors themselves noted that for several methods, evaluation from later work could attain a higher accuracy, which shows that a small difference in the evaluation experiment setting could make a difference.

In summary, I lean towards rejecting this work.

**Questions:**

1. How do you select $\alpha$ in your experiments?
2. Why do you interpolate between the empirical and certified bounds? Are you following the prior work by De Palma et al.? What does this trick buy us?

---

> ### Author Response · Authors · 2025-11-23
>
> We sincerely thank Reviewer S6Q6 for their time and feedback, which helps us improve the quality of the submission.
>
> We will address the raised points and questions individually. Equation numbers refer to the new paper revision, where requested changes are highlighted in blue.
> We sincerely hope the below addresses the reviewer's concerns, and remain eagerly available for further discussion and potential clarifications.
>
> > *I feel overwhelmed by the notations in this manuscript*
>
> Thank you for your feedback on this: in order to help the reader, the newly-added appendix F provides an explanation on the employed notation, which we hope the reviewer will find helpful.
> While we acknowledge that the notation employed in the manuscript may be relatively heavy, we believe that it is necessary to precisely express the employed quantities.
> Indeed, we are happy that both reviewer 1hxv (*the manuscript is well written*) and reviewer DdPg (*The paper is well-written*) positively evaluated the quality of the writing.
>
> > *I also don't see why Lemma 3.3 is important.*
>
> Lemma 3.3 writes the CC-IBP convex combinations as a function of the feature-space convex combinations within the distillation loss, for the case where *both convex combinations share the same $\alpha$ coefficient*.
> This provides a strong coupling between the proposed distillation loss and the CC-IBP loss, motivating the use of the same convex combination coefficient for both, and hence removing an hyper-parameter.
>
> > *The motivation of this work is to give certified robust models the same standard accuracy as adversarially trained models [...] I am not even sure if the central problem studied in this work is a valid one.*
>
> Our work stems from the observation that trade-offs between empirical robustness and standard performance are significantly better than those between certified robustness and standard performance.
> We show that distilling from adversarially-trained teachers can significantly improve on both standard accuracy and certified robustness, leading to a new state-of-the-art in the area, and we ascribe this to the inherent robustness of the adversarial teacher's representation.
> In order to demonstrate the importance of a robust teacher representation, **the newly-added appendix E.10 shows that distillation from clean teachers results in markedly worse performance profiles**.
>
> A discussion on the effect of knowledge distillation is provided in Appendix E.1.
> The high-level technical intuition behind the distillation training process is as follows:
> - The goal of robust training is to learn using loss the loss in eq. (3), which is however intractable.
> - Adversarial training replaces eq. (3) by a lower bound: eq. (4). This means using cross-entropy on the adversarial logit differences. The use of this loss results in fairly good trade-offs between standard accuracy and empirical robustness, but low certified accuracy.
> - CC-IBP uses cross-entropy on the CC-IBP logit differences defined in eq. (7), which include a verifiability-inducing term, yielding good certified accuracy at the expense of standard accuracy and empirical robustness compared to adversarial training.
> - The distillation term defined in Proposition 3.2 pushes the student model to learn a feature-space representation (by acting on the convex combinations to mirror the behaviour of CC-IBP as per Lemma 3.3) mirroring that of the adversarially-trained teacher, hence transferring some of its superior clean accuracy and empirical robustness onto the student.
> - The presence of the CC-IBP term in eq. (10) enforces verifiability onto the distilled representation, converting part of the superior empirical robustness into an increase in certified accuracy.
>
> Concerning the *validity of the central problem*, training networks for certified robustness to adversarial examples: the importance of this research direction within the adversarial ML community is testified by, among other things, the large number of papers in the area published in major ML conferences.
> Examples include: SortNet (NeurIPS22, oral), SABR (ICLR23, notable top 25%), TAPS (NeurIPS23), CC-IBP (ICLR24), CTBench (ICML25).

---

> > ### Author Response · Authors · 2025-11-23
> >
> > > *The reported results are not significantly better*
> >
> > The presented improvements are in line with some of the best recent papers in the area of $l_{\infty}$ ReLU-based certified training.
> > For reference purposes, we here report the improvements upon the best-performing previous method of relevant previous works in the area (compared to both literature results and re-tuned baselines in each paper, when available), focusing on benchmarks where ReLU-based certified training is the state-of-the-art.
> >
> > CIFAR-10 2/255
> > | Work | Venue | Std. Acc. Impr. [%] | Cert. Acc. Impr. [%] |
> > | -------- | ------- | -------- | ------- |
> > | SortNet | NeurIPS22 | -10.68 | -3.56 |
> > | SABR | ICLR23 | 1.05 | 0.87 |
> > | STAPS | NeurIPS23 | 0.52 | 0.14 |
> > | CC-IBP | ICLR24 | 0.20 | 0.50 |
> > | *CC-Dist* | *submission* | *1.46* | *0.82* |
> >
> > TinyImageNet 1/255
> > | Work | Venue | Std. Acc. Impr. [%] | Cert. Acc. Impr. [%] |
> > | -------- | ------- | -------- | ------- |
> > | SortNet | NeurIPS22 | -0.02 | 0.54 |
> > | SABR | ICLR23 | 2.93 | 2.59 |
> > | STAPS | NeurIPS23 | 0.13 | 1.70 |
> > | Exp-IBP | ICLR24 | 0.03 | 0.33 |
> > | *CC-Dist* | *submission* | *2.5* | *1.35* |
> >
> > ImageNet64 1/255
> > | Work | Venue | Std. Acc. Impr. [%] | Cert. Acc. Impr. [%] |
> > | -------- | ------- | -------- | ------- |
> > | SortNet | NeurIPS22 | -1.44 | 0.81 |
> > | Exp-IBP | ICLR24 | 2.40 | 0.91 |
> > | *CC-Dist* | *submission* | *2.23* | *0.27* |
> >
> > > *I cannot find in the paper how $\alpha$ is selected in the experiments*
> >
> > Extensive details concerning the $\alpha$ selection process are provided in appendix D.2.4.
> > In summary: we use the same $\alpha$ value from CC-IBP in De Palma et al. (2024b) on CIFAR-10 for both CC-IBP and CC-Dist.
> > On TinyImageNet and ImageNet64, we found that the performance profile of CC-IBP can be strictly improved by decreasing $\alpha$ compared to De Palma et al. (2024b).
> > CC-Dist uses the same $\alpha$ value as CC-IBP in all benchmarks except on TinyImageNet, where we found a larger $\alpha$ to result in simultaneously better standard
> > performance and certified robustness compared to all results from the literature. Nevertheless, as reported in appendix D.2.4, re-using the same $\alpha$ on CIFAR-10 significantly improves upon CC-IBP.
> >
> > > *unclear whether the improvement of the proposed method comes from a different evaluation setting.*
> >
> > Owing to the cost of certified training and of post-training verification via branch-and-bound, reporting literature results is the de facto standard in the area (De Palma et al., 2024b; Mao et al., 2023; Müller et al., 2023).
> > Nevertheless, we would like to stress that we re-evaluate CC-IBP, our main baseline and a state-of-the-art expressive loss from De Palma et al. (2024b) under the same evaluation setting, hence decoupling the performance improvement associated to our proposed distillation technique from the evaluation itself.
> > We furthermore point out that CTBench (Mao et al., 2025), an independent experimental study under a unified setting, confirmed the state-of-the-art performance of expressive losses in the area.
> >
> > > *Why do you interpolate between the empirical and certified bounds? Are you following the prior work by De Palma et al.? What does this trick buy us?*
> >
> > Both the worst-case distillation loss in equation (8) and the worst-case classification loss in equation (2) are intractable to compute.
> > As a result, we approximate them through three different quantities:
> > 1. A lower bound computed by evaluating (2) or (8) on a concrete input point determined by adversarial attacks;
> > 2. An upper bound computed as the worst-case for (2) or (8) over a tractable convex network relaxation (over-approximation);
> > 3. Interpolations between lower and upper bounds obtained through convex combinations, which allow to cover a continuous range of trade-offs between lower and upper bounds.
> > As shown by De Palma et al. (2024b), the ability to smoothly interpolate between lower and upper bounds (termed *expressivity*, see definition 3.1 in the relative work) is crucial to attain state-of-the-art performance in the area, and we here build on that result, which was empirically confirmed by the later CTBench (Mao et al., 2025). Empirically, this is linked to the ability to precisely tune the amount of bounds regularization depending on the specific benchmark.

---

### Official Review · Reviewer_LhHm · 2025-10-31

**Soundness:** 3
**Presentation:** 3
**Contribution:** 2
**Rating:** 6
**Confidence:** 3

**Summary:**

This paper seeks to bridge the gap between empirical and certifiable robustness by distilling knowledge from adversarially trained teachers into certifiably trained students through CC-Dist, a feature-space distillation objective tightly coupled with an expressive certified training loss, CC-IBP. The method forms convex combinations of adversarial features and interval bound propagation (IBP) feature bounds, demonstrating that the resulting distillation objective upper-bounds the worst-case feature-level risk and varies continuously and monotonically with the parameter α. This same parameter coherently regulates both the certified loss and the distillation target. Implemented with auto_LiRPA and verified post hoc using OVAL with α–β–CROWN, CC-Dist consistently improves both standard and certified accuracies over pure CC-IBP on CIFAR-10, TinyImageNet, and ImageNet64, achieving state-of-the-art trade-offs for ReLU networks. Teacher–student analyses further show that while teachers exhibit higher standard and empirical robustness but negligible certifiable guarantees, CC-Dist students achieve strong certification with competitive accuracy.

**Strengths:**

1. This work achieves a tight theoretical coupling between feature-space distillation and expressive certifiable training (CC-IBP). The distillation target is constructed as a convex combination of adversarial features and their IBP lower and upper bounds, and the authors prove that the distillation term upper-bounds the worst-case feature-level risk while varying continuously and monotonically with the interpolation parameter α. Under the affine classifier assumption, α jointly regulates both the certified loss and the distillation target, thereby unifying empirical discriminability and certifiable stability in a single coherent and interpretable framework.

2. The proposed method exhibits consistent improvements across multiple vision benchmarks and perturbation radii, and teacher–student comparisons reveal a clear division of strengths. Teachers demonstrate higher standard and empirical robustness but weak certifiability, whereas students achieve significantly stronger certificates while maintaining competitive accuracy. Moreover, curve analyses of distillation strength show stable trends in training dynamics and performance metrics as hyperparameters vary, supporting the overall soundness of the design.

3. The method also demonstrates strong generality and practical deployability. The two endpoint cases of distillation (α = 0 / 1) remain effective in several settings and can be readily transferred to other certified training methods (e.g., SABR). The implementation is built on widely used community toolchains, facilitating reproducibility and extension. The paper further notes that under large perturbation settings, specialized 1-Lipschitz architectures offer superior performance, suggesting a promising direction for extending CC-Dist to such structures.

**Weaknesses:**

1. In scenarios with large perturbation radii, such as ε = 8/255 on CIFAR-10, the ReLU-based CC-Dist has yet to surpass specialized 1-Lipschitz architectures such as SortNet. This suggests that, although the approach advances the accuracy–certificate frontier in most settings, it has not fundamentally bridged the structural gap in large-ε regimes. Moreover, both theoretical and experimental analyses mainly focus on ReLU activations and IBP, while the adaptation and potential benefits across other activation functions or architectures, such as strictly Lipschitz-constrained networks, remain to be systematically validated.

2. The method relies on high-quality adversarial teachers, which introduces additional training costs. Achieving certifiable guarantees also requires substantial computation and time on branch-and-bound verifiers. The paper reports that the increased certified accuracy of CC‑Dist models is accompanied by higher verification runtime. Due to computational cost, teacher certification is evaluated only on a subset of the test set, suggesting that the overall training and verification pipeline remains expensive, particularly for larger datasets and more complex models.

3. The study focuses primarily on ℓ∞ threat models and visual classification benchmarks, with limited evaluation under other paradigms such as ℓ₂ or semantic-level perturbations, or in tasks such as detection and segmentation. Although the shared parameter α between the distillation and the certified loss is theoretically motivated, its sensitivity and robustness across datasets, architectures, and training schedules are not comprehensively analyzed, and β is mostly assigned a fixed empirical value. Furthermore, the affine-head assumption improves interpretability but limits applicability to more complex nonlinear or attention-based heads, leaving open questions regarding broader generalization and complete ablation.

**Questions:**

1. It would be valuable to include preliminary results or a clear plan for extending CC-Dist to 1-Lipschitz architectures (e.g., SortNet) under the ε = 8/255 setting, in order to better assess whether the observed performance gap primarily stems from methodological factors or architectural differences.
2. A more systematic ablation of α/β and teacher strength across datasets, architectures, and multiple random seeds would help quantify the sensitivity of the method to hyperparameters and teacher quality, thereby reinforcing the robustness and generality of the conclusions.
3. Could you quantify how tight the IBP feature bounds used in the distillation loss are (e.g., per‑layer bound gaps vs. approximate worst‑case features) and whether looseness concentrates in deeper layers?

---

> ### Author Response · Authors · 2025-11-23
>
> We sincerely thank Reviewer LhHm for their positive evaluation and feedback, which helps us improve the quality of the work.
>
> We are glad that the reviewer appreciated the consistent improvements of distillation across the considered benchmarks, and the generality of the approach.
> We now address the raised weaknesses and questions individually. Requested changes are highlighted in blue in the revision.
>
> > *the adaptation and potential benefits across other activation functions or architectures [...] remain to be systematically validated.*
>
> Following the relevant literature (De Palma et al., 2024b; Mao et al., 2023; Müller et al., 2023; Shi et al., 2021), we here focused on CNN7 owing to its state-of-the-art performance on $\ell_{\infty}$ benchmarks. However, we believe that the idea of distilling from adversarially-trained teachers may generalize well beyond the considered context.
> In order to support this, the newly-added appendix E.8 demonstrates that CC-Dist successfully improves trade-offs between standard and certified accuracies on (i) a modified CNN7 with hyperbolic tangent (tanh) activations, (ii) an 18-layer residual network common in the adversarial training literature (PreActResNet18).
> We believe these results validate the broader generality of the proposed idea, and pave the way for exciting further work in the area.
>
> > *the overall training and verification pipeline remains expensive*
>
> The large cost associated to certified training and post-training verification via branch-and-bound is, unfortunately, common to the entire research area, and understood to be the price for formal guarantees.
> As visible from appendix E.7, the training overhead associated to our distillation scheme comes from the teacher training costs and is relatively low on CIFAR-10, and around 40% and 70% on ImageNet64 and TinyImageNet, respectively.
> The increase in post-training verification time (which, in any case, generally dominates the training time) instead results from the increased IBP loss associated to the distillation process.
> We believe this overhead to be proportionate to the presented performance improvements, and leave the investigation of more efficient distillation schemes for future work.
>
> > *limited evaluation under other paradigms such as $\ell_2$ or semantic-level perturbations, or in tasks such as detection and segmentation*
>
> While, as common in the relevant certified training literature (De Palma et al., 2024b; Mao et al., 2023; Müller et al., 2023), we here focus on vision classification benchmarks and $\ell_\infty$ perturbations, we believe that adapting the proposed approach to other tasks and perturbation types is an exciting direction for future work, and thank the reviewer for the remark. We are confident that the distillation process will be beneficial beyond classification tasks and $\ell_\infty$ perturbations.
>
> > *parameter $\alpha$ [...] its sensitivity and robustness across datasets, architectures, and training schedules are not comprehensively analyzed, and $\beta$ is mostly assigned a fixed empirical value*
>
> We were here interested in investigating the effect of distillation on $\alpha$ values leading to state-of-the-art performance, in order to improve upon the best-known trade-offs between certified and standard accuracy. We refer the reviewer to Figure 1 from De Palma et al. (2024b) for a detailed analysis of the effect of varying $\alpha$ coefficients for CC-IBP.
> $\beta$ is mostly assigned a fixed value, which is found to yield strong performance in all of our main results, in order to avoid the introduction of an additional hyper-parameter.
>
> > *the affine-head assumption [...] limits applicability*
>
> While the affine-head assumption is required for Lemma 3.3, the distillation process can be applied for non-affine classification heads.
> Nevertheless, as we show in the newly-added appendix E.8, distilling on the last affine layer yields strictly better robustness-accuracy trade-offs (compared to the penultimate layer).
> We do not hence believe that the assumption limits applicability.

---

> ### Author Response · Authors · 2025-11-23
>
> > *extending CC-Dist to 1-Lipschitz architectures (e.g., SortNet) under the ε = 8/255 setting*
>
> The results in appendix E.8 point to the wider applicability of the proposed distillation scheme.
> Indeed, investigating the benefits of distillation for Lipschitz networks is an exciting direction for future work.
> Extra care would have to be taken when training effective teacher models for the purpose, as we found a large degree of similarity between the teacher and the student to be required for the distillation process to be effective. This may be harder to achieve when distilling onto SortNet, for instance, owing to its custom activation function.
> Ad hoc teachers may have to be defined and employed.
> We are nevertheless hopeful that distillation would be beneficial: encouraging evidence in this sense is provided by the work on learning certifiably-robust perceptual similarity metrics through distillation discussed in related work (Ghazanfari et al., 2024).
>
> > *multiple random seeds [...] sensitivity of the method to hyperparameters and teacher quality*
>
> As shown by tables 8 and 13 and mentioned above, the teacher needs to be carefully chosen in order for the distillation process to be effective.
> A sensitivity study for the $\beta$ hyper-parameter is provided in figure 1.
> Appendix E.4 provides details on experimental variability for multiple seeds on CIFAR-10, demonstrating that the improvements in robustness-accuracy trade-offs are statistically significant on the two relative settings.
>
> > *Could you quantify how tight the IBP feature bounds used in the distillation loss are (e.g., per‑layer bound gaps vs. approximate worst‑case features) and whether looseness concentrates in deeper layers?*
>
> Generally speaking, the IBP bounds are indeed fairly loose, and will get looser with the depth of the network.
> For instance, while the IBP accuracies for CC-Dist and CC-IBP are both 0% on CIFAR-10 2/255 and TinyImageNet, using the best bounds between CROWN and IBP (CROWN/IBP) results in certified accuracies above 50% and above 20% for both methods on CIFAR-10 2/255 and TinyImageNet, respectively.
> This is, however, dependent on the benchmark and on the employed $\alpha$ value: CROWN/IBP certified accuracy coincides with IBP accuracy for both CC-Dist and CC-IBP on CIFAR-10 8/255.
> As noted in the introduction, the prevalence of IBP bounds in certified training is due to its favourable optimisation properties (Jovanovic et al., 2022; Lee et al., 2021).

---

### Official Review · Reviewer_1hxv · 2025-11-06

**Soundness:** 3
**Presentation:** 3
**Contribution:** 3
**Rating:** 6
**Confidence:** 4

**Summary:**

This paper proposes CC-Dist, a certified training methods for provable robustness. It augments CC-IBP with a feature-space knowledge distillation objective to achieve a better trade-off between the clean accuracy and the provable robust accuracy.

**Strengths:**

++ I generally like the idea: although adversarial training will not generally obtain provably robust models, these may not be problem of adversarial training but the current verifier cannot certify their robustness. By incorporating the adversarially trained models into provable robust learning in a smooth way can make the best of both sides.

++ The algorithm is straightforward and the manuscript is well written: compared with CC-IBP, it only adds one single hyper-parameter which is not very sensitive, making the proposed CC-Dist easy to adopt on top of existing algorithms.

++ The experiments are relatively comprehensive and convincing.

**Weaknesses:**

1. The gaps between the proposed method and baselines are relatively small on Table 1, so running the experiments for multiple times and report the performance variance would be better.

2. PGD-40 is utilised as the metric for empirical robustness. However, the most reliable empirical robustness evaluation scheme is AutoAttack in RobustBench. It would be better to use AutoAttack instead.

3. The scope is a bit limited, as the results and discussions are demonstrated for $l_\infty$ bounded perturbations only and based on the architecture CNN-7 with ReLU as the only activation functions. I am not sure if the same method can be generalizable to deeper neural networks, activation functions and perturbation types.

In general, I think this work is a meaningful contribution to the community. It can be improved if the concerns above can be properly addressed.

Minor: there are some additional related literature:

* AutoAttack as the reliable empirical evaluation metric: "Reliable evaluation of adversarial robustness with an ensemble of diverse parameter-free attacks" (2020)

* When discussing linear network relaxations for provable robustness, there is a series of works utilising geometric properties to derive the loss function: "Provable robustness of relu networks via maximization of linear regions." (2019), "Training Provably Robust Models
by Polyhedral Envelope Regularization" (2023)

* Randomized smoothing is also an important series of provable robustness: "Certified Adversarial Robustness via Randomized Smoothing" (2019)

**Questions:**

First some questions about the weakness:

1. What is the variance of the performance? Is it big? In addition, please use AutoAttack as the metric for empirical robustness for a more reliable evaluation.

2. Is this method extendable to boarder architectures, other activation functions and perturbations beyond $l_\infty$ bounded ones?

In addition, I have additional questions:

1. In Appendix D.2.4, the author mentioned "tuning was carried our on the evaluation sets." Isn't it unfair?

2. Why do you choose the features before final affine layer for alignment in knowledge distillation? Can you choose an earlier layer? You can still use IBP or CROWN-IBP to derive the output bound to calculate the original loss $^{CC}\mathcal{L}^{\mathcal{C}_\epsilon}_{f_\theta}$.

---

> ### Author Response · Authors · 2025-11-23
>
> We sincerely thank Reviewer 1hxv for their valuable questions and suggestions, which help us increase the quality of the work, and for their positive evaluation.
>
> We are particularly glad that the reviewer liked the idea behind our approach and found our experiments to be comprehensive and convincing.
>
> We now individually address the reported weaknesses and the questions. Requested changes are highlighted in blue in the revision.
>
> > *The gaps between the proposed method and baselines are relatively small on Table 1*
>
> While the improvements may seem small, we believe they are in line with some of the best recent papers in the area of $l_{\infty}$ ReLU-based certified training.
> For reference purposes, we here report the improvements upon the best-performing previous method of relevant previous works in the area (compared to both literature results and re-tuned baselines in each paper, when available), focusing on benchmarks where ReLU-based certified training is the state-of-the-art.
>
> CIFAR-10 2/255
> | Work | Venue | Std. Acc. Impr. [%] | Cert. Acc. Impr. [%] |
> | -------- | ------- | -------- | ------- |
> | SortNet | NeurIPS22 | -10.68 | -3.56 |
> | SABR | ICLR23 | 1.05 | 0.87 |
> | STAPS | NeurIPS23 | 0.52 | 0.14 |
> | CC-IBP | ICLR24 | 0.20 | 0.50 |
> | *CC-Dist* | *submission* | *1.46* | *0.82* |
>
> TinyImageNet 1/255
> | Work | Venue | Std. Acc. Impr. [%] | Cert. Acc. Impr. [%] |
> | -------- | ------- | -------- | ------- |
> | SortNet | NeurIPS22 | -0.02 | 0.54 |
> | SABR | ICLR23 | 2.93 | 2.59 |
> | STAPS | NeurIPS23 | 0.13 | 1.70 |
> | Exp-IBP | ICLR24 | 0.03 | 0.33 |
> | *CC-Dist* | *submission* | *2.5* | *1.35* |
>
> ImageNet64 1/255
> | Work | Venue | Std. Acc. Impr. [%] | Cert. Acc. Impr. [%] |
> | -------- | ------- | -------- | ------- |
> | SortNet | NeurIPS22 | -1.44 | 0.81 |
> | Exp-IBP | ICLR24 | 2.40 | 0.91 |
> | *CC-Dist* | *submission* | *2.23* | *0.27* |
>
> > *I am not sure if the same method can be generalizable to deeper neural networks, activation functions and perturbation types.*
>
> We thank the reviewer for the remark.
> While we focused on CNN7 owing to its state-of-the-art performance on $\ell_{\infty}$ benchmarks and prevalence in the literature (De Palma et al., 2024b; Mao et al., 2023; Müller et al., 2023; Shi et al., 2021), we believe that the idea of distilling from adversarially-trained teachers may generalize well beyond the considered context.
> In order to support this, the newly-added Appendix E.8 demonstrates that CC-Dist successfully improves trade-offs between standard and certified accuracies on (i) a modified CNN7 with tanh activations, (ii) an 18-layer residual network common in the adversarial training literature.
> We believe that our distillation process may be beneficial to other perturbation and model types. In particular, we believe the investigation of distillation for Lipschitz networks, which are the state-of-the-art on $\ell_2$ perturbations and on the $\ell_{\infty}$ 8/255 CIFAR-10 benchmark, to be an exciting direction for future work.
>
> > *additional related literature*
>
> We sincerely thank the reviewer for the pointers: we had already cited randomized smoothing, and have now included the other provided references.
>
> > *What is the variance of the performance? Is it big?*
>
> Owing to the large cost of repeated training and verification, and in line with previous work  (De Palma et al., 2024b; Mao et al., 2023; Müller et al., 2023; Mao et al., 2024a; 2025) (see reproducibility statement), we provide single-seed results for the main experiments.
> However, experimental variability on CIFAR-10 is provided in appendix E.4, demonstrating that the improvements in robustness-accuracy trade-offs are statistically significant on the two relative settings.
>
> > *please use AutoAttack as the metric for empirical robustness for a more reliable evaluation.*
>
> We thank the reviewer for suggesting this: Tables 2 and 4 have been edited to report AutoAttack accuracy instead of PGD-40 accuracy. The evaluation confirms the trends visible from the PGD-40 accuracy.
>
> > *In Appendix D.2.4, the author mentioned "tuning was carried our on the evaluation sets." Isn't it unfair?*
>
> As stated in Appendix D.2.4, a large body of previous work (Gowal et al., 2019; Zhang et al., 2020; Shi et al., 2021; Müller et al., 2023; De Palma et al., 2024b; Mao et al., 2025) carries out tuning directly on the evaluation set.
> Doing the same hence ensures a fair comparison with these results.
> While we agree this is not a great practice, it became widely established in the area owing to the cost of post-training verification.
> Appendix G.5 from De Palma et al. (2024b) presents a CC-IBP validation study on CIFAR-10, showing that this practice does not significantly influence the relative performance across expressive losses.

---

> > ### Author Response · Authors · 2025-11-23
> >
> > > *Why do you choose the features before final affine layer for alignment in knowledge distillation? Can you choose an earlier layer?*
> >
> > Thank you for this question: while in principle any layer can be chosen for the distillation, we focused on the last layer as we found this to result in better performance.
> > The newly-added Appendix E.9 shows the effect of distillation on earlier layers, focusing on the penultimate affine layer.
> > As shown in table 11, distilling on the last affine layer yields strictly better robustness-accuracy trade-offs.
> > Given that distilling on earlier layers implies that a larger portion of the network is exclusively trained using the CC-IBP loss, we ascribe the inferior performance to insufficient teacher-student coupling.

---

> > > ### Comment · Reviewer_1hxv · 2025-11-25
> > >
> > > I thank the authors for further clarification in the rebuttal, which has addressed most of my concerns. I believe this work is beneficial for the community and may inspire more subsequent works.
> > >
> > > I decide to keep my original positive rating and vote for an acceptance. I did not further increase my rating because the improvement in the experiment part is a bit marginal and further improvement may be possible. (while I acknowledge that some baselines are very strong, and it is not an easy task).

---

### Author Response · Authors · 2025-11-23
**General response**

We sincerely thank all reviewers for their time and feedback, which helped us improve the quality of the submission.

We are happy that reviewers 1hxv and DdPg positively evaluated the quality of the writing (*"the manuscript is well written"*, *"The paper is well-written"*), appreciated the main idea behind our work (*"I generally like the idea"*, *"The idea is clean and effective"*), and its coupling with CC-IBP (*"making the proposed CC-Dist easy to adopt on top of existing algorithms"*, *"The proposed algorithm is efficient and integrates nicely into one of the SOTA certified training algorithm"*).
Furthermore, we are glad that the experimental evaluation was positively received by reviewers 1hxv, LhHm and DdPg (*"The experiments are relatively comprehensive and convincing"*, *"The proposed method exhibits consistent improvements across multiple vision benchmarks and perturbation radii"*, *"The improvement on the clean accuracy is significant, with nontrivial improvement on the certified robustness"*).

Our work shows that knowledge distillation from adversarially-trained teachers improves the state-of-the-art in $\ell_\infty$ certified training for ReLU networks.
At the same time, as highlighted by the discussion with the reviewers, we believe that distillation holds great potential for the wider certified training literature, and the new revision contains new experiments pointing to its generality.
Appendix E.3 demonstrates that distillation is applicable and beneficial to MTL-IBP and IBP-R (in addition to SABR, already presented at submission).
Furthermore, the newly-added appendix E.8 shows that CC-Dist improves trade-offs between certified and standard accuracy also for a different activation function and a relatively deep residual network.
Finally, appendix E.10 demonstrates that a robust teacher representation is fundamental for the overall success of the distillation process, shedding further light on the presented scheme.

We hope the revision and our responses address the reviewers' concerns, and remain eager to exchange with reviewers further.

---

### Meta-Review · Area_Chair_8DSz · 2026-01-05

**Summary:**

The paper proposes CC-Dist, a method designed to improve the trade-off between standard accuracy and certified robustness by distilling knowledge from empirically robust (adversarially trained) teachers into certified students. While the integration of distillation with the CC-IBP loss was appreciated by some reviewers (1hxv, DdPg) for its intuitive appeal, I recommend rejecting this submission. This decision is primarily driven by significant concerns regarding the paper's presentation and the significance of the empirical results. Reviewer S6Q6 raised fundamental issues with the manuscript's readability, citing "overwhelming" and confusing notation that obscures the core ideas. Furthermore, while the method achieves state-of-the-art results on specific benchmarks, Reviewers 1hxv and LhHm noted that the performance gains are relatively marginal and come at the expense of increased training and verification costs. Consequently, the paper currently falls short of the bar for acceptance due to clarity issues and the limited impact of the proposed improvements relative to the added complexity.

**Reviewer Concerns:**

**Addressed Concerns:**
- The authors successfully addressed specific technical questions raised during the rebuttal. Notably, they adopted AutoAttack for more reliable empirical evaluation (addressing Reviewer 1hxv), provided ablation studies on feature selection (addressing Reviewer 1hxv), and demonstrated the method's potential applicability to other architectures and activation functions (addressing Reviewers 1hxv and LhHm).

**Outstanding Concerns:**
The most critical outstanding issues relate to the Presentation and Significance of the work.

- Presentation & Clarity: Reviewer S6Q6 found the notation to be unnecessarily heavy and confusing, arguing that it hinders the understanding of what is fundamentally a simple idea. Despite the authors adding an appendix to explain the notation, the core manuscript remains difficult to parse, which is a significant barrier to publication.

- Motivation: Reviewer S6Q6 questioned the underlying motivation, specifically why interpolating bounds and distilling from a non-certified teacher guarantees better certified performance. The intuitive link remains weak for some readers.

- Marginal Gains vs. Cost: Reviewers 1hxv and LhHm pointed out that the improvements over baselines are marginal. Reviewer LhHm specifically noted that for larger perturbation radii, the method does not bridge the gap to specialized architectures like SortNet, yet it introduces significant computational overhead for teacher training and student verification.

**Reviewer Scores:**

Reviewer S6Q6: I believe their score would remain low (2). Their critique focused on fundamental flaws in presentation and motivation that were not fully resolvable through rebuttal appendices alone.

Reviewer 1hxv & Reviewer LhHm: I believe these reviewers might have lowered their scores (e.g., to 5 or 4) had they engaged more deeply with the concerns raised by Reviewer S6Q6. While they gave "Marginally above threshold" (6) scores, both highlighted that the results were "marginal" or "expensive." A discussion on the trade-off between the complexity of the notation/method and the actual performance gain would likely have dampened their enthusiasm.

Reviewer DdPg: Their score would likely remain high, as they were strongly convinced by the "clean idea" and efficiency, though I find their assessment perhaps overlooks the presentation hurdles identified by others.

---

### Decision · Program_Chairs · 2026-01-26

Reject